# MAF1, a repressor of RNA polymerase III-dependent transcription, regulates bone mass

**Ellen Phillips[1], Naseer Ahmad[2], Li Sun[2], James Iben[3], Christopher J Walkey[1], Aleksandra Rusin[1], Tony Yuen[2], Clifford J Rosen[4], Ian M Willis[5], Mone Zaidi[2], Deborah L Johnson[1]***

[1]Department of Molecular and Cellular Biology, Baylor College of Medicine, Houston, United States; [2]Departments of Medicine and Pharmacological Sciences and Center for translational Medicine and Pharmacology, Icahn School of Medicine at Mount Sinai, New York, United States; [3]Molecular Genomics Core, Eunice Kennedy Shriver National Institute of Child Health and Human Development, National Institutes of Health, Bethesda, United States; [4]Center for Clinical and Translational Research, Maine Medical Center Research Institute, Scarborough, United States; [5]Departments of Biochemistry and Systems and Computational Biology, Albert Einstein College of Medicine, Bronx, United States

*For correspondence:
deborah.johnson@bcm.edu

**Abstract** MAF1, a key repressor of RNA polymerase (pol) III-mediated transcription, has been shown to promote mesoderm formation in vitro. Here, we show that MAF1 plays a critical role in regulating osteoblast differentiation and bone mass. Global deletion of *MAF1* (*Maf1*-/- mice) produced a high bone mass phenotype. However, osteoblasts isolated from *Maf1*-/- mice showed reduced osteoblastogenesis ex vivo. Therefore, we determined the phenotype of mice overexpressing MAF1 in cells from the mesenchymal lineage (*Prx1*-Cre;LSL-*MAF1* mice). These mice showed increased bone mass. Ex vivo, cells from these mice showed enhanced osteoblastogenesis concordant with their high bone mass phenotype. Thus, the high bone mass phenotype in *Maf1*-/- mice is likely due to confounding effects from the global absence of MAF1. MAF1 overexpression promoted osteoblast differentiation of ST2 cells while MAF1 downregulation inhibited differentiation, indicating MAF1 enhances osteoblast formation. However, other perturbations used to repress RNA pol III transcription, inhibited osteoblast differentiation. However, decreasing RNA pol III transcription through these perturbations enhanced adipogenesis in ST2 cells. RNA-seq analyzed the basis for these opposing actions on osteoblast differentiation. The different modalities used to perturb RNA pol III transcription resulted in distinct gene expression changes, indicating that this transcription process is highly sensitive and triggers diverse gene expression programs and phenotypic outcomes. Specifically, MAF1 induced genes known to promote osteoblast differentiation. Furthermore, genes that are induced during osteoblast differentiation displayed codon bias. Together, these results reveal a novel role for MAF1 and RNA pol III-mediated transcription in osteoblast fate determination, differentiation, and bone mass regulation.

## Editor's evaluation

In this manuscript, Phillips et al., have used several complementary in vivo and in vitro approaches to analyze the effects of regulated MAF1 expression or inhibition of RNA pol III transcription on osteogenesis and adipocyte differentiation. The data are well controlled and of excellent quality, providing novel insights into Maf1 and RNA polymerase-mediated transcriptions in skeleton biology.

## Introduction

RNA polymerase (pol) III transcribes various untranslated RNAs including 5S rRNA and tRNAs. RNA pol III-derived transcripts play essential roles in several processes, including protein synthesis and secretion (*Dieci et al., 2013*; *Dieci et al., 2007*). In addition to RNA pol III, transcription of tRNAs requires the recruitment of TFIIIC and TFIIIB to the promoter. TFIIIB consists of Brf1, TATA-binding protein (TBP) and B-double prime (*Orioli et al., 2012*). RNA pol III-dependent transcription is also tightly regulated through either direct or indirect mechanisms that control TFIIIB recruitment to the promotor (*Gomez-Roman et al., 2003*; *Kenneth et al., 2007*; *Felton-Edkins et al., 2003*; *Sriskanthadevan-Pirahas et al., 2018*; *Crighton et al., 2003*; *Sutcliffe et al., 2000*; *White et al., 1996*; *Woiwode et al., 2008*). MAF1 is a key repressor of RNA pol III-dependent transcription ( *Graczyk et al., 2015*; *Johnson et al., 2007*). It binds to RNA pol III and prevents the interaction between RNA pol III and TFIIIB (*Vorländer et al., 2019*; *Vannini et al., 2010*). MAF1 has also been shown to regulate a variety of RNA pol II-transcribed targets (*Johnson et al., 2007*; *Palian et al., 2014*; *Khanna et al., 2014*; *Li et al., 2016*; *Lee et al., 2015*), and acts as a tumor suppressor (*Palian et al., 2014*; *Li et al., 2016*), regulates metabolism (*Willis et al., 2018*; *Bonhoure et al., 2015*), and longevity (*Shetty et al., 2020*; *Cai and Wei, 2016*).

RNA pol III-mediated transcription has been shown to play an important role during development and cellular differentiation. Two RNA pol III isoforms are differentially expressed between pluripotent and differentiated embryonic stem cells (ESCs) (*Haurie et al., 2010*; *Wong et al., 2011*; *Wang et al., 2020*). RNA pol III-dependent transcription also modulates the formation of hematopoietic lineages in zebrafish (*Wei et al., 2016*) and transcription is downregulated during skeletal muscle differentiation of *Xenopus tropicalis* (*McQueen et al., 2019*). MAF1 enhances the formation of mesoderm in embryonic stem cells, and the downregulation of RNA pol III-dependent transcription enhances the differentiation of ESCs and 3T3-L1 cells into adipocytes (*Chen et al., 2018*).

Diseases associated with ribosomal disfunctions, ribosomopathies, are commonly associated with bone marrow, skeletal and craniofacial disorders (*Trainor and Merrill, 2014*). This surprising tissue specificity suggests that these tissues may be particularly sensitive to alterations in protein synthesis. For example, Treacher Collins syndrome is caused by mutations in *POLR1C*, *POLR1D,* or *TCOF1*, which affect rDNA transcription by RNA pol I and ribosome biogenesis (*Noack Watt et al., 2016*; *Dauwerse et al., 2011*). RNA pol III-derived transcripts may also play a role as POLR1C and POLR1D are common subunits of both RNA pol I and RNA pol III. In yeast, Treacher Collins syndrome-related mutations in *POLR1D* result in altered functions of both RNA pol I and III (*Walker-Kopp et al., 2017*). Furthermore, RNA pol III-dependent transcription may play a role in cerebellofaciodental syndrome which is associated with mutations in the RNA pol III-specific transcription factor, Brf1. This syndrome is characterized by a neurodevelopmental phenotype as well as changes in the facial and dental structure and delayed bone age (*Borck et al., 2015*; *Jee et al., 2017*; *Honjo et al., 2021*). Additionally, bone phenotypes have been described in a subgroup of patients with mutations in RNA pol III subunits (*Borck et al., 2015*; *Jee et al., 2017*; *Honjo et al., 2021*; *Terhal et al., 2020*; *Ghoumid et al., 2017*).

It is known that MAF1 regulates mesoderm formation and adipocyte differentiation (*Chen et al., 2018*). Osteoblasts and adipocytes are both derived from the mesenchymal lineage (*Chen et al., 2016*) and mutations in *BRF1* and RNA pol III subunits are associated with bone-related phenotypes. These facts led us to hypothesize that regulation of RNA pol III-dependent transcription by MAF1 may play a fundamental role in osteoblast differentiation, bone formation and hence, bone mass. Here, we show that both whole body deletion of *MAF1* in mice and tissue-specific overexpression of MAF1 in stromal cells of the long bones enhances bone mass in vivo, while MAF1 induces osteoblast differentiation in vitro. To further examine the role of MAF1 and RNA pol III-mediated transcription in osteoblast differentiation, we attempted to study the effect on osteoblast differentiation by repressing transcription through different approaches. Surprisingly, while MAF1 induced osteoblast differentiation, repression of RNA pol III-dependent transcription, by either chemical inhibition of RNA pol III or by Brf1 knockdown, decreased osteoblast differentiation. Thus, changes in MAF1 expression produce an opposing effect on osteoblast differentiation compared with other approaches that repress RNA pol III-mediated transcription. We further show that these three different approaches to decrease RNA pol III-dependent transcription result in divergent gene expression changes. Altered MAF1 expression affects the expression of the osteoblast differentiation gene program. Together, these findings reveal

that MAF1 plays a key role in osteoblast differentiation and bone mass regulation and that its ability to regulate RNA pol III-dependent transcription contributes to the observed phenotypic outcomes.

## Results

## MAF1 overexpression stimulates osteoblast lineage cells to differentiate into mature osteoblasts, and enhances adipogenesis

To determine the role of MAF1 on bone mass and bone formation in vivo, we examined the bone phenotype of the global *Maf1-/-* mouse model (*Bonhoure et al., 2015*). Micro-computed tomography (μCT) was used to determine femur, tibia, and spine bone volume and microstructural parameters in mature male mice at 12 weeks of age. When compared to age-matched wild-type (WT) mice, *Maf1-/-* mice showed a significant increase in bone volume, trabecular number, and trabecular thickness in the spine and increased bone volume and trabecular thickness in the tibia. Femur samples showed a similar trend without reaching statistical significance (*Figure 1—figure supplement 1*). To further determine the mechanism of this increase in bone mass, we performed histomorphometric analysis (*Figure 1—figure supplement 2*). This showed that bone formation parameters, mineralizing surface, mineral apposition rate, and bone formation rate, were significantly increased in the spine. Tibiae showed an increase in mineralization surface. Overall, these data suggest that the increase in bone mass in *Maf1-/-* mice is due to increased bone formation. To determine changes at the cellular level, we isolated primary bone marrow stromal cells and hematopoietic cells from these mice to determine their ex vivo capacity to form osteoblasts and osteoclasts, respectively. Surprisingly, osteoblast formation was reduced, while osteoclastogenesis was increased in cells derived from *Maf1-/-* mice (*Figure 1—figure supplement 3*). This suggests that MAF1 increases osteoblast and decreases osteoclast formation in long-term ex vivo cultures. However, this change does not reflect the increase in bone volume seen in *Maf1-/-* mice. Because ex vivo cultures are not affected by signals originating from external tissues, we hypothesized that the increase in bone formation in *Maf1-/-* mice is likely the result of non-cell-autonomous effects arising from the deletion of *MAF1* in other tissues. The ex vivo data indicate that MAF1 is a positive regulator of osteoblast differentiation, and its overexpression in osteoblasts would therefore be expected to increase osteoblastogenesis.

We determined the effect of MAF1 overexpression specifically in the long bones by developing a transgenic mouse strain with an HA-tagged *MAF1* construct inserted in the *Rosa26* locus. The expression of MAF1 was driven by a hybrid cytomegalovirus enhancer chicken β-actin (CAGGS) promoter with a lox-stop-lox cassette inserted between the promoter and *MAF1*. This *Rosa-lox-stop-lox-MAF1-HA* (LSL-MAF1) strain was then crossed to a *Prx1*-Cre mouse to overexpress MAF1 in the mesenchyme of the developing limb bud (*Logan et al., 2002*). HA expression in the femur following Cre-recombination was confirmed by western blotting. qRT-PCR showed a ~16 fold increase in MAF1 transgene expression compared to endogenous *MAF1* mRNA (*Figure 1A–B*). No gross phenotypic changes were observed, and the weight of the mice at 12 weeks was unchanged (*Figure 1C*). To assess the effect of increased MAF1 expression on bone mass, we employed μCT analysis on femurs of 12-week-old male mice. MAF1 overexpression in *Prx1*-Cre; LSL-MAF1 mice led to an increase in bone volume, trabecular number and thickness, and connectivity density, and a corresponding reduction in trabecular separation when compared with Cre-controls. Cortical thickness was not significantly increased (*Figure 1D*). Histomorphometric analysis confirmed an increase in trabecular number and a reduction in trabecular separation, while other parameters were not significantly affected (*Figure 1—figure supplement 4*).

To determine the effect of MAF1 overexpression at the cellular level, primary stromal cells were isolated from femurs and tibia of *Prx1*-Cre; *LSL-MAF1* mice and cultured ex vivo. These primary cells displayed an increase in MAF1 expression and showed a corresponding decrease in tRNA gene transcription (*Figure 1E*). When MAF1-overexpressing stromal cells were allowed to differentiate into bone-forming osteoblasts in media containing ascorbic acid and β-glycerolphosphate, there was a clear increase in their mineralizing capacity seen by alizarin red staining (*Figure 1F*). These results suggest that MAF1 enhances the differentiation or function of osteoblasts. We found no change in the osteoclastogenic cytokines receptor activator of NF-κβ ligand (RANKL) and osteoprotegerin (OPG) in the femurs, indicating that changes in the OPG/RANKL ratio did not play a role (*Figure 1G*).

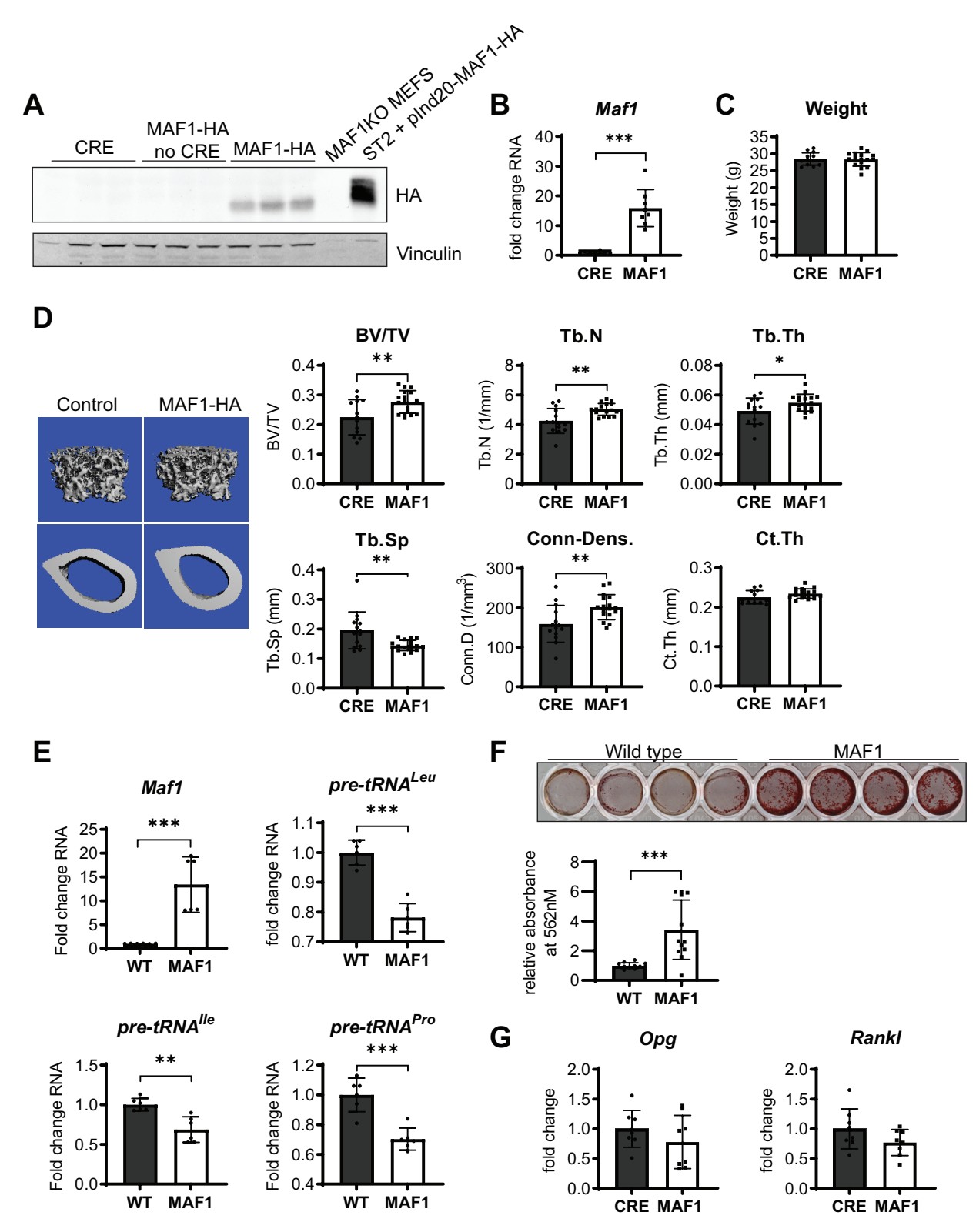

**Figure 1.** Bone-specific overexpression of MAF1-HA increases bone volume in mice. (**A**) Western blot of HA expression in the femur of 12-week-old male Prx1-Cre MAF1-HA mice compared to Prx1-Cre-WT and WT-MAF1-HA mice. (**B**) qRT-PCR analysis showing MAF1 RNA in femurs from Prx1-Cre-MAF1 mice and control Prx1-Cre mice (n=8). (**C**) Weights in gram of 12-week-old Prx1-Cre or Prx1-Cre-MAF1 mice. (**D**) Left, representative images of μCT of femoral bone. Right, quantification of μCT analysis: bone volume/total volume (BV/TV), trabecular number (Tb.N), trabecular thickness (Tb.Th),

*Figure 1 continued on next page*

*Figure 1 continued*

trabecular separation (Tb.Sp), connectivity density (Conn-Dens.), and cortical thickness (Ct.Th). n=13 for Prx1-Cre and n=17 for MAF1 mice. (**E**) qRT-PCR of MAF1 and pre-tRNAs in primary stromal cells isolated from 6- to 8-week old WT or MAF1 overexpressing mice (n=6). (**F**) Representative plate of Alizarin red-labeled mineralization of WT and MAF1-HA primary stromal cells (top). Quantification of Alizarin red after destaining with 10% CPC. (**G**) qRT-PCR of Opg and Rankl in Prx1-Cre and MAF1 overexpressing femurs at 12 weeks (n=8). Results represent means ± SD, *p<0.05, **p<0.01, ***p<0.001 determined by Student's t-test. *Figure 1—source data 1* contains uncropped images of western blots.

The online version of this article includes the following source data and figure supplement(s) for figure 1:

**Source data 1.** Immunoblot analysis 1.

**Figure supplement 1.** *Maf1⁻/⁻* mice show increased bone mass in the spine.

**Figure supplement 2.** *Maf1⁻/⁻* mice show increased bone formation in the spine.

**Figure supplement 3.** Ex vivo analysis *Maf1⁻/⁻* cells show decreased osteoblast differentiation and increased osteoclast formation.

**Figure supplement 4.** Histomorphometric analysis of Prx1-Cre-MAF1-HA mice.

To further confirm a role for MAF1 on osteoblast differentiation, we overexpressed MAF1 in the mouse stromal cell line ST2 using a doxycycline (Dox)-inducible MAF1-HA construct. Cells stably expressing either the MAF1-HA construct or a control vector were treated with 1 µM Dox and differentiated into osteoblasts. Ectopic MAF1 expression was confirmed by western blot (*Figure 2A*) and qRT-PCR and resulted in the reduction of pre-tRNA^Ile and pre-tRNA^Leu gene transcription (*Figure 2B*). MAF1 overexpression resulted in enhanced staining for alkaline phosphatase (Alp), a marker for early osteoblast differentiation (*Figure 2C*), as well as increased in vitro mineralization as noted on alizarin red staining (*Figure 2D*). MAF1 overexpression also resulted in a significant increase in the expression of osteoblast marker genes, namely collagen type 1 alpha (*Col1A*), *Sp7*, *osteocalcin*, *Alp,* and bone sialoprotein (*Bsp*) (*Figure 2E*).

In parallel, loss of function studies, MAF1 expression was decreased in ST2 cells using two different shRNAs (*Figure 3A*). As expected, this resulted in an increase in pre-tRNA expression, particularly on day 10 (*Figure 3B*). *MAF1* knockdown resulted in a decrease in Alp staining (*Figure 3C*) and a robust reduction in mineralization (*Figure 3D*). Osteoblast markers, namely *Sp7*, *Alp,* and *Bsp* were significantly downregulated in these cells (*Figure 3E*).

Collectively, these results indicate that MAF1 functions to promote osteoblast differentiation. Reduced expression of MAF1 in ST2 cells and in bone marrow stromal cells derived from *Maf1⁻/⁻* mice resulted in a decrease in osteoblast differentiation. MAF1 overexpression in ST2 cells increased osteoblast differentiation while MAF1 overexpression, specifically in the mesenchymal cells of the long bones, increased bone mass. The paradoxical phenotype in *Maf1⁻/⁻* mice therefore likely resulted from yet uncharacterized, non-cell-autonomous confounding effects on osteoblasts arising from global *MAF1* deletion.

Finally, MAF1 has been shown to promote adipogenesis since knockdown of *MAF1* in pre-adipocytes reduces adipocyte formation (*Chen et al., 2018*). We therefore examined how alterations in MAF1 expression may affect the differentiation of ST2 cells into adipocytes. We found that MAF1 overexpression produced an increase in Oil red O stained cells and upregulated the expression of adipogenesis genes *Pparg*, *Cebpa* and *Fabp4* (*Figure 2—figure supplement 1*). To determine if adipocyte formation was affected *MAF1* deficient mice, we isolated primary cells from *Maf1⁻/⁻* mice. These cells showed increased tRNA transcription as expected and showed decreased differentiation into adipocytes as seen by Oil Red O stain (*Figure 3—figure supplement 1A, B*). Consistent with these results, histological analysis of femurs from 12-week-old *Maf1⁻/⁻* mice showed that both adipocyte number and adipocyte volume was reduced (*Figure 3—figure supplement 1C*). These results confirm that in addition to promoting osteogenesis, MAF1 enhances adipocyte differentiation.

## MAF1-independent approaches to repress RNA pol III-dependent transcription decrease osteoblast differentiation

As MAF1 functions as a repressor of RNA pol III-dependent transcription (*Johnson et al., 2007*; *Orioli et al., 2016*), we determined if other approaches that inhibit RNA pol III-dependent transcription would produce a similar increase in osteoblast differentiation. ST2 cells were treated with ML-60218, a chemical inhibitor of RNA pol III (*Wu et al., 2003*) or with DMSO vehicle as the control. Cells were treated for three days, starting one day before the addition of differentiation media. Two days after the

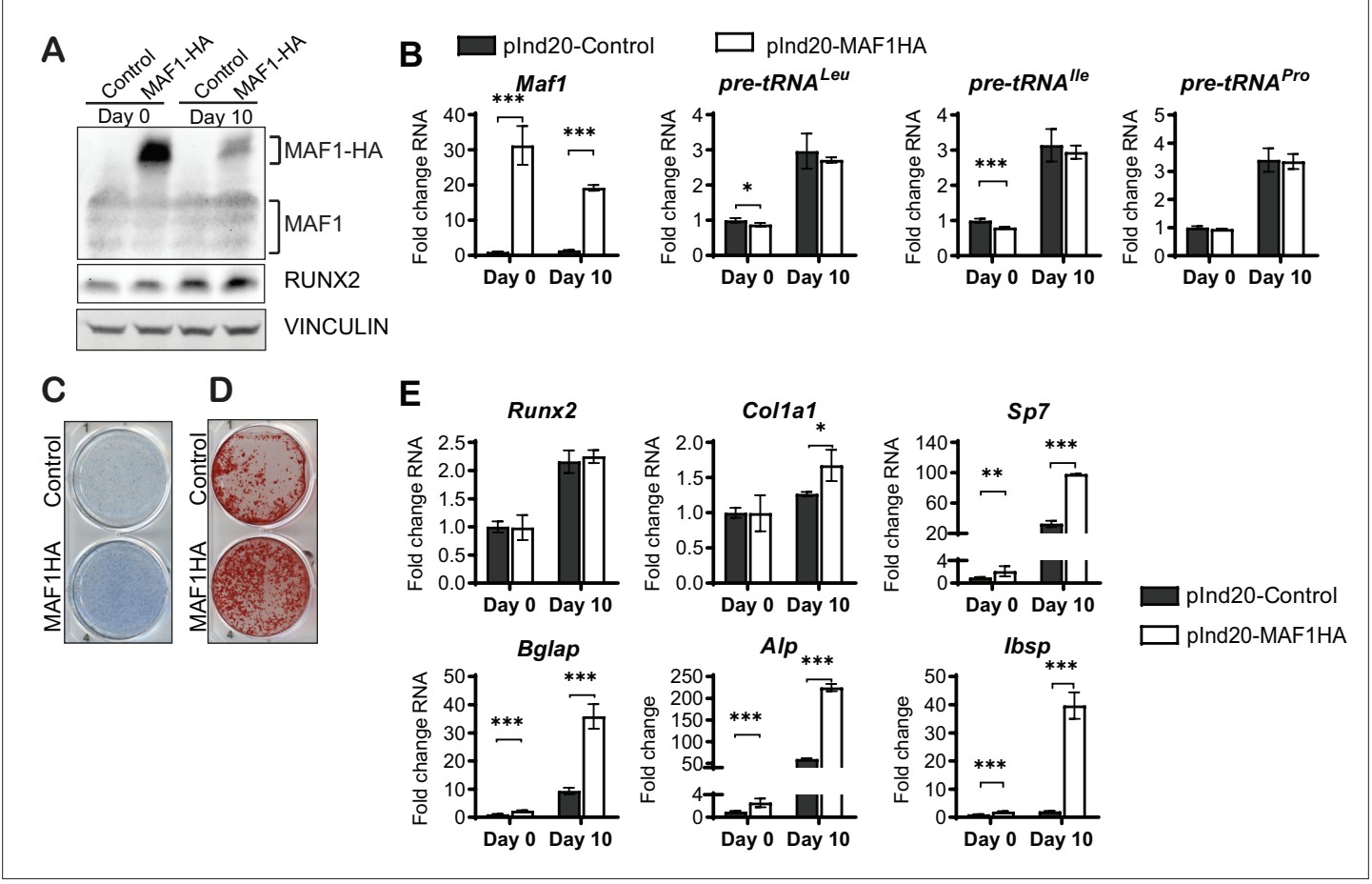

**Figure 2.** MAF1 increases in vitro osteoblast differentiation and mineralization. ST2 cells were infected with a doxycycline (Dox)-inducible pInd20-MAF1HA or control construct. Cells were treated with 1 μM Dox starting 1 day before differentiation was started. (**A**) Western blot analysis showing MAF1, Runx2, and Vinculin in ST2 cells differentiated into osteoblast on day 0 and day 10. (**B**) qRT-PCR analysis showing MAF1 and pre-tRNA expression in ST2 cells pre- and during osteoblast differentiation. (**C**) Representative image of alkaline phosphatase (Alp) staining of control and MAF1-HA expressing cells. (**D**) Representative image of alizarin red analysis of ST2 cells overexpressing control or MAF1-HA after culture in osteoblast differentiation medium. (**E**) qRT-PCR analysis showing relative expression of Runx2, Col1α, Sp7 (Osterix), Alp, and Bone sialoprotein before and 10 days after the addition of osteoblast differentiation medium. Results represent means ± SD of three independent replicates, *$p<0.05$, **$p<0.01$, ***$p<0.001$ determined by Student's t-test with Holm correction. *Figure 2—source data 1* contains uncropped western blot images, *Figure 2—source data 2* contains uncropped images of stained plates.

The online version of this article includes the following source data and figure supplement(s) for figure 2:

**Source data 1.** Immunoblot analysis 2.

**Source data 2.** Immunoblot analysis 2.

**Figure supplement 1.** MAF1 overexpression enhances adipogenesis in ST2 cells.

**Figure supplement 1—source data 1.** Immunoblot analysis 3.

**Figure supplement 1—source data 2.** Immunoblot analysis and differentiation assays.

initiation of differentiation, ML-60218 was removed, and cells were allowed to differentiate without further manipulation. tRNA gene transcription was significantly reduced by ML-60218 treatment (*Figure 4A*). However, in contrast to what was observed with MAF1 overexpression, reduction of RNA pol III transcription by ML-60218 resulted in a decrease in Alp and Alizarin red staining (*Figure 4B and C*) and a significant reduction in the expression of osteoblast marker genes (*Figure 4D*). This effect was not specific to ST2 cells, as the differentiation of primary stromal cells derived from C57BL/6 mice was also significantly reduced by ML-60218 treatment (*Figure 4—figure supplement 1*).

Using a complementary approach, we downregulated the expression of the RNA pol III-specific transcription factor *Brf1* to reduce RNA pol III transcription (*Figure 5A and B*). Similar to ML-60218

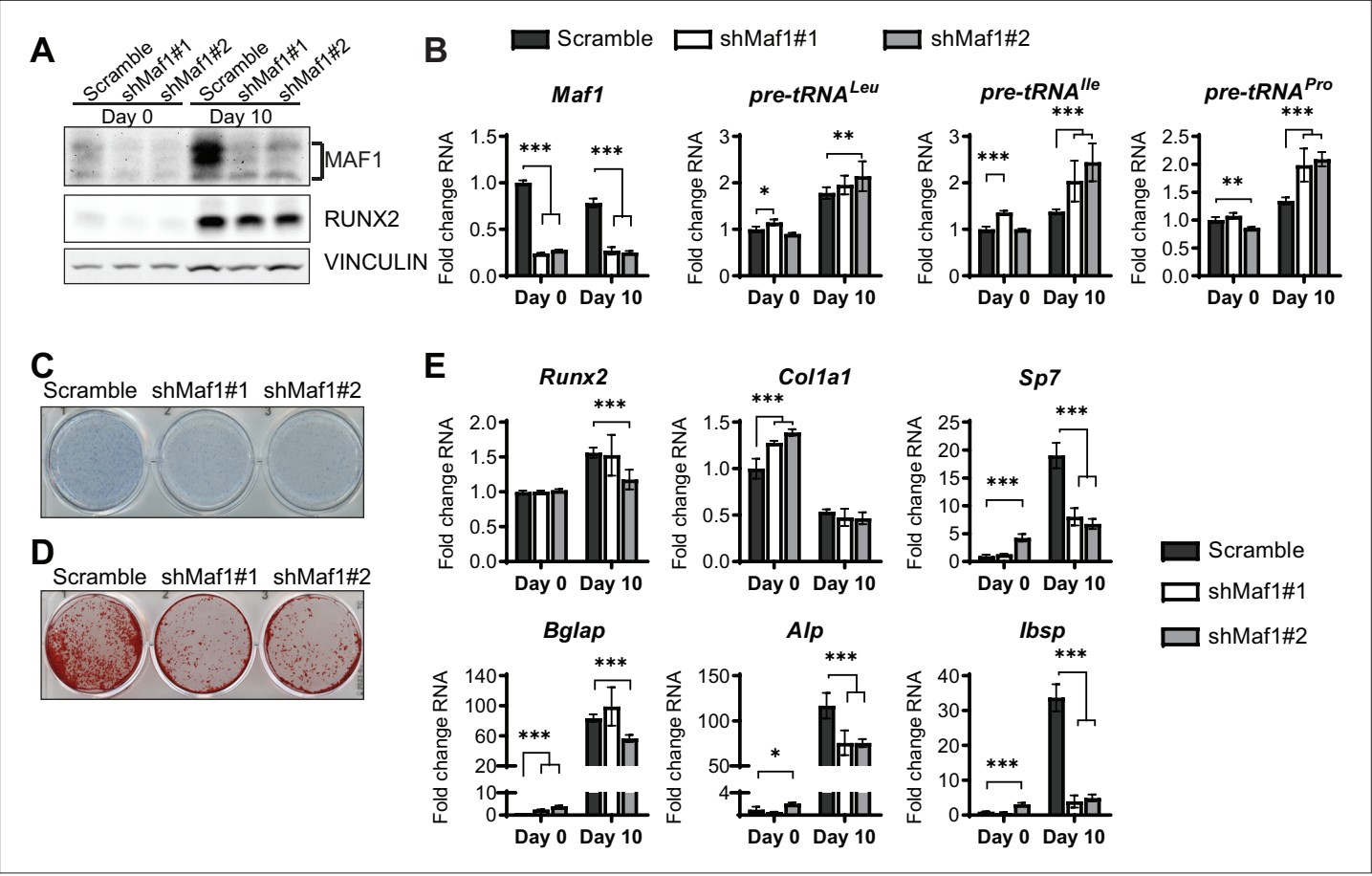

**Figure 3.** MAF1 knockdown decreases osteoblast differentiation of ST2 cells. (**A**) Western blot analysis showing MAF1, Runx2, and Vinculin expression in cells infected with a Scramble construct or MAF1 shRNA before, or 10 days after adding osteoblast differentiation medium. (**B**) qRT-PCR analysis of MAF1 and pre-tRNAs of ST2 cells expressing Scramble of shMaf1 before and on day after adding osteoblast differentiation medium. (**C**) Alkaline phosphatase staining of ST2 cells expressing scramble or lentiviral MAF1 shRNA after culture in osteoblast differentiation medium. (**D**) Alizarin red analysis of cells with scramble or MAF1 shRNA after culture in osteoblast differentiation medium. (**E**) qRT-PCR analysis showing relative expression of Runx2, Col1α, Sp7, Alp, and Bone sialoprotein before, and 10 days after addition of osteoblast differentiation medium. Results represent means ± SD of three independent replicates, *p<0.05, **p<0.01, ***p<0.001 determined by Student's t-test with Holm correction. *Figure 3—source data 1* contains uncropped western blot images, *Figure 3—source data 2* contains uncropped images of stained plates.

The online version of this article includes the following source data and figure supplement(s) for figure 3:

**Source data 1.** Adipogenesis assays.

**Source data 2.** Immunoblot analysis 3.

**Figure supplement 1.** MAF1 deficiency decreases adipocyte differentiation in vitro and bone marrow adipocytes in vivo.

**Figure supplement 1—source data 1.** Adipogenesis assay 2.

treatment, *Brf1* knockdown decreased Alp and Alizarin red staining, as well as osteoblast marker expression (*Figure 5C–E*). These results indicate that, while different approaches to decrease RNA pol III-dependent transcription all affect osteoblast differentiation, increased MAF1 expression promotes differentiation, while *Brf1* knockdown and ML-60218 treatment repress osteoblast differentiation.

To further delve into the opposing effects of MAF1 overexpression versus Brf1 downregulation or ML-60218 treatment, we examined the relative effects of these three perturbations on adipocyte differentiation from ST2 cells. MAF1-independent approaches to repress RNA pol III-dependent transcription produced an increase in adipogenesis observed upon Oil red O staining and adipocyte marker expression (*Figure 4—figure supplement 2*, *Figure 5—figure supplement 1*). Interestingly, this was similar to what was observed by MAF1 overexpression. Thus, while all three mechanisms of inhibiting RNA pol III transcription (ML-60218 treatment, *Brf1* knockdown and *MAF1* overexpression) enhance adipocyte differentiation, only MAF1 overexpression enhances osteoblast differentiation.

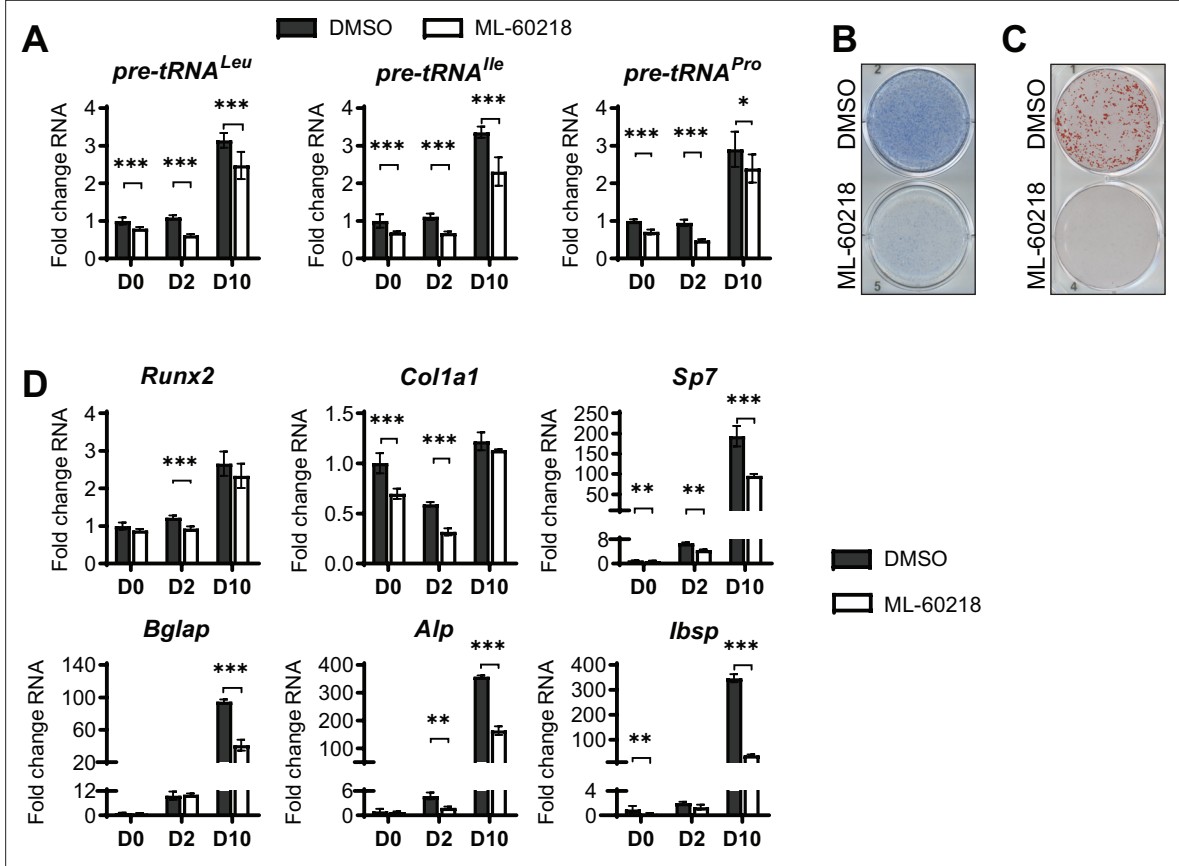

**Figure 4.** inhibition of RNA pol III-dependent transcription by ML-60218 decreases osteoblast differentiation and mineralization. ST2 cells were treated with 40 µM ML-60218 for 3 days, starting on day –1 and differentiated into osteoblasts by addition of osteoblast differentiation medium on day 0. (**A**) qRT-PCR analysis of pre-tRNAs before and during differentiation after ML-60218 or DMSO treatment of ST2 cells. (**B**) Representative image of alkaline phosphatase (Alp) staining of ST2 cells after osteoblast differentiation in DMSO or ML60218 treated cells. (**C**) Representative image of alizarin red analysis of ST2 cells after osteoblast differentiation and ML-60218 or DMSO treatment. (**D**) qRT-PCR analysis of Runx2, Col1α, Sp7, Osteocalcin, Alp and Bone Sialoprotein in ST2 cells on day 0, day 2 and day 10 during osteoblast differentiation. Results represent means ± SD of three independent replicates. *p<0.05, **p<0.01, ***p<0.001 determined by Student's t-test with Holm correction. *Figure 4—source data 1* contains uncropped images of stained plates.

The online version of this article includes the following source data and figure supplement(s) for figure 4:

**Source data 1.** Osteoblast and adipocyte assays.

**Figure supplement 1.** ML-60216 treatment decreases osteoblast differentiation of primary stromal cells.

**Figure supplement 1—source data 1.** Immunoblot analysis 4.

**Figure supplement 2.** ML-60218 treatment enhances adipogenesis of ST2 cells.

**Figure supplement 2—source data 1.** Immunoblot assays and differentiation assays.

**Figure supplement 2—source data 2.** Immunoblot analysis 4.

## RNA sequencing shows that different perturbations to alter RNA pol III transcription result in distinct gene expression profiles

To investigate the contrasting effect of altering MAF1 expression *versus* ML-60218 treatment or Brf1 knockdown on osteoblast differentiation, we performed RNA-seq on ST2 cells harvested before the start of differentiation, day 0, and on day 4. Each manipulation was compared to their own appropriate controls and triplicates were analyzed for each datapoint. Genes with an adjusted p-value <0.05 and a log2fold change >0.7 in either direction were considered. Each intervention used to manipulate RNA pol III-dependent transcription resulted in distinct changes in gene expression on both day 0 and day 4 (*Figure 6A–B*). There was little overlap in gene expression changes caused by each approach used to manipulate RNA pol III transcription. To explore whether these changes relate

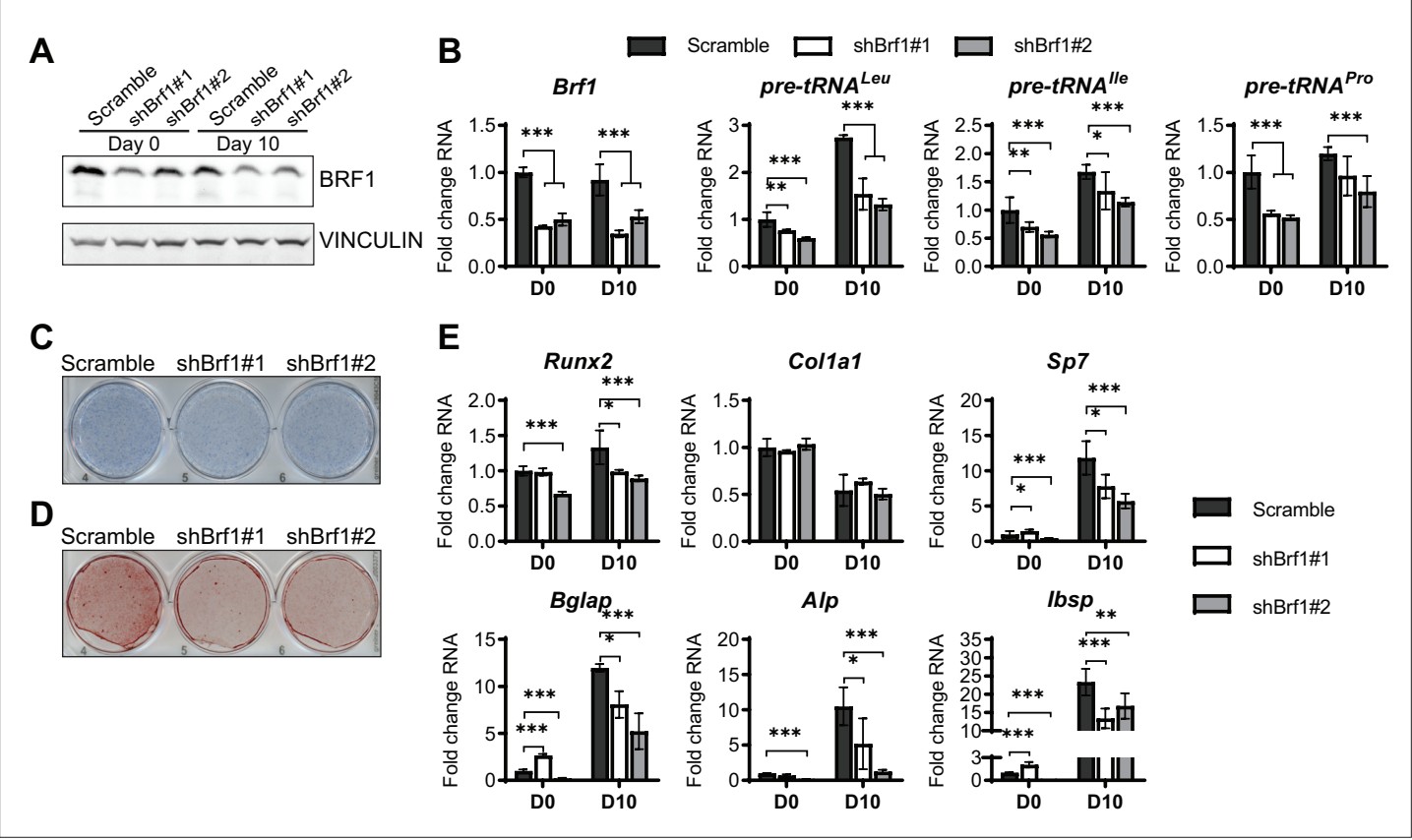

**Figure 5.** Inhibition of RNA pol III-dependent transcription Brf1 knockdown decreases osteoblast differentiation and mineralization. ST2 cells were stably infected with scramble or Brf1 shRNA lentivirus and differentiated into osteoblasts by addition of osteoblast differentiation medium on day 0. (**A**) Western blot analysis showing Brf1 and Vinculin expression in cells infected with a scramble construct Brf1 shRNA before or 10 days after adding osteoblast differentiation medium. (**B**) qRT-PCR analysis of Brf1 and pre-tRNAs of ST2 cells expressing Scramble of shBrf1 before and on day after adding osteoblast differentiation medium. (**C**) Representative image of alkaline phosphatase (Alp) staining of ST2 cells expressing scramble or lentiviral Brf1 shRNA after culture in osteoblast differentiation medium. (**D**) Representative image of alizarin red analysis of cells with Scramble or Brf1 shRNA after culture in osteoblast differentiation medium. (**E**) qRT-PCR analysis showing relative expression of Runx2, Col1α, Sp7 (Osterix), Alp, and Bone sialoprotein before and 10 days after the addition of osteoblast differentiation medium. Results represent means ± SD of three independent replicates, *p<0.05, **p<0.01, ***p<0.001 determined by Student's t-test with Holm correction. *Figure 5—source data 1* contains uncropped western blot images, *Figure 5—source data 2* contains uncropped images of stained plates.

The online version of this article includes the following source data and figure supplement(s) for figure 5:

**Source data 1.** Immunoblot analysis and adipocyte differentiation assays.

**Source data 2.** Differentiation analysis 1.

**Figure supplement 1.** Brf1 knockdown enhances adipogenesis in ST2 cells.

**Figure supplement 1—source data 1.** GEO data from RNA sequencing analysis.

**Figure supplement 1—source data 2.** GEO dataset analysis.

to specific biological processes, we performed gene ontology (GO) enrichment analysis on the data at day 0. Comparing the 20 most significantly enriched subgroups, we found that MAF1 overexpression and *Brf1* knockdown resulted in enrichment for GO terms previously shown to relate to the regulation osteoblast differentiation, such as extracellular matrix organization and ossification. In contrast, ML-60218 treatment enriched mostly for lipid metabolism and adipocyte differentiation genes. The enrichment observed for these GO terms, without inducing differentiation, suggests that manipulating RNA pol III-dependent transcription positions cells in a manner that affects lineage determination. Our GO analysis also uncovered other biological processes that were changed. However, these varied between the subgroups (*Figure 6—figure supplement 1*, *Figure 6—figure supplement 2*,

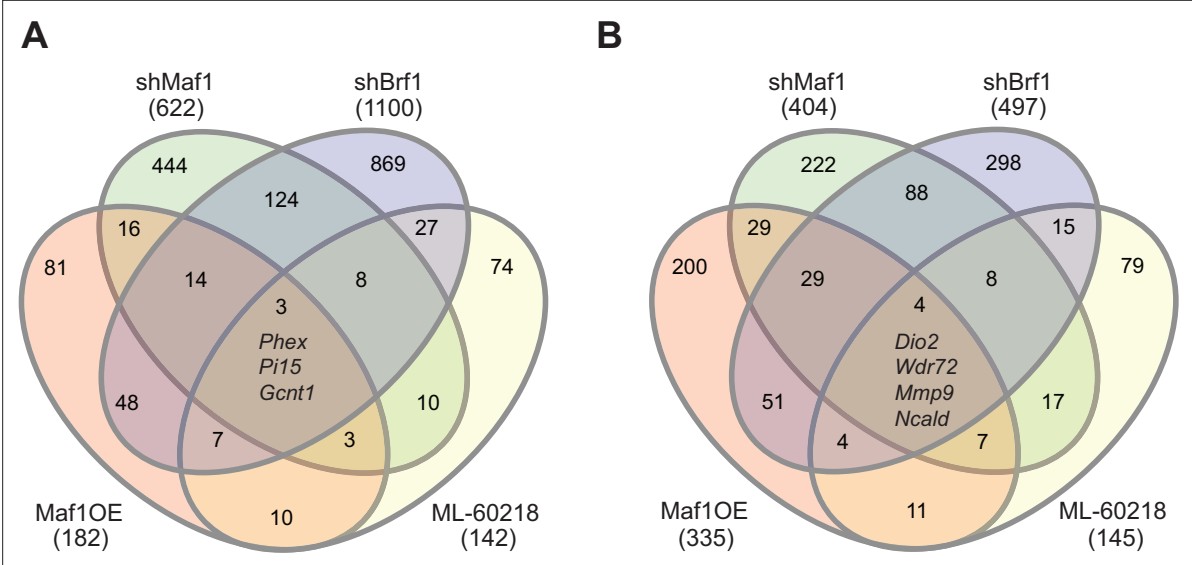

**Figure 6.** Manipulating RNA pol III in different manners results in distinct gene pools. Changes in gene expression were determined by padj<0.05 and foldchange >|log2 0.7|. Venn diagram showing overlap in gene changes (either increased or decreased) on day 0 (**A**) or (**B**) day 4 (**B**). Genes that were changed in all groups are denoted. MAF1OE genes changes between pInd20-MAF1 and Pind20-Control; shMAF1 was compared to scramble control, shBrf1 was compared to scramble control; ML-60218 was compared to DMSO control. *Figure 6—source data 1* contains excel files with all differentially expressed genes.

The online version of this article includes the following source data and figure supplement(s) for figure 6:

**Source data 1.** GEO analysis from RNA sequencing analysis 2.

**Figure supplement 1.** MAF1 overexpression results enrichment for terms related to bone biology.

**Figure supplement 2.** MAF1 knockdown causes enrichment for terms related to bone and renal biology.

**Figure supplement 3.** ML-60218 treatment results in enrichment in gene ontology (GO) terms related to lipid metabolism.

**Figure supplement 4.** Brf1 knockdown produces gene changes that are enriched in gene ontology (GO) terms related to bone biology and immune responses.

**Figure supplement 5.** Genes altered by changes in MAF1 expression prior to differentiation.

*Figure 6—figure supplement 3*, *Figure 6—figure supplement 4*). Overall, these results reveal that different approaches to manipulate RNA pol III can produce disparate gene expression changes that lead to different biological outcomes.

To determine which genes were specifically altered by MAF1, we compared changes in gene expression on day 0 by MAF1 overexpression and *MAF1* knockdown. We only considered genes that were at least log2fold 0.7 changed in an opposing direction in each treatment set (*Figure 6—figure supplement 3*). This uncovered several genes that were not changed in the same direction by *Brf1* knockdown or ML-60218 treatment which have known effects on bone. Among these were phosphate-regulating endopeptidase homolog, X linked (Phex), which increased upon MAF1 overexpression and decreased by *Brf1* knockdown or ML-60218 treatment. Of note, *Phex* plays a key role in promoting bone mineralization and phosphate homeostasis (*Rowe, 2012*). In addition, *Col15a1*, which is associated with early osteoblast differentiation (*Lisignoli et al., 2017*), and *Lysyl oxidase like 2* (*Loxl2*), which is involved in collagen crosslinking (*Mitra et al., 2019*), were also increased by MAF1 overexpression. In contrast, *Rhomboid 5 homolog 2* (*Rhbdf2*) expression was decreased. *Rhbdf2* knockout mice display a high bone mass phenotype (*Levy et al., 2020*), suggesting RHBDF2 may play a role in regulating bone mass. Together, the results suggest that MAF1 may specifically regulate a subset of genes that play a role in regulating bone mass.

To identify a potential mechanism, we further compared changes in gene expression that occurred during osteoblast differentiation, without manipulating RNA pol III-dependent transcription. We examined potential changes in codon usage during osteoblast differentiation by comparing codon usage of upregulated genes with an exhaustive list of gene-coding sequences in mice. This revealed a significant bias towards the use of certain codons during osteoblast differentiation (*Figure 7*). Codon usage

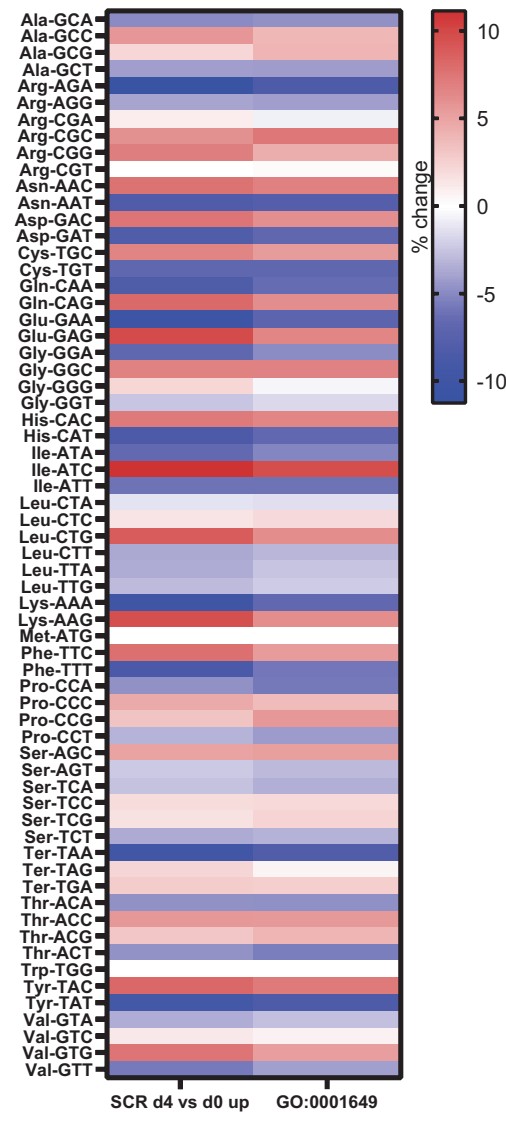

**Figure 7.** Genes expressed during osteoblast differentiation display significant codon bias. Relative changes in codon usage during osteoblast differentiation day 4, compared to day 0 for SCR control cells (left) or of genes that are members of the GO term 0001649 (osteoblast differentiation) (right). *Figure 7—source data 1* contains excel files with all codon analysis.

The online version of this article includes the following source data for figure 7:

**Source data 1.** GEO dataset analysis.

in our dataset was predominantly overlapping to codon bias in genes belonging to the GO term for osteoblast differentiation (GO:0001649), showing similar codon bias in two independent datasets. This suggests, for the first time, that codon bias may play a role during osteoblast differentiation.

## Discussion

MAF1 is a key repressor of transcription by RNA pol III (*Johnson et al., 2007*; *Vorländer et al., 2019*). Changes in MAF1 expression have been shown to enhance adipocyte differentiation (*Chen et al., 2018*; *Figure 2—figure supplement 1* and *Figure 3—figure supplement 1*). *Maf1*[-/-] mice are shorter than WT mice and exhibit a lean phenotype with resistance to diet-induced obesity, decreased fertility and fecundity, and increased longevity/healthspan (*Bonhoure et al., 2015*). Here, we demonstrate that these mice display an increase in bone volume and bone formation. However, *Maf1*[-/-] derived primary stromal cells showed a decrease in osteoblast formation in ex vivo cultures. While it is possible this is due to different culture conditions, this latter finding was consistent with the effects of MAF1 overexpression specifically in the mesenchyme of long bones, which resulted in enhanced osteoblast differentiation and an increase in bone mass. In these latter mice, compared with *Maf1*[-/-] mice, any confounding actions due to the absence of MAF1 in other tissues, such as possible endocrine or paracrine effects, are limited. Thus, our results indicate that the ability of MAF1 to regulate bone mass involves both cell autonomous and non-cell-autonomous actions. This idea is further corroborated by our in vitro results showing that osteoblast differentiation is regulated by increasing or decreasing MAF1 expression in ST2 cells, suggesting that MAF1 promotes osteoblast differentiation and mineralization. Our findings are summarized in *Table 1*. As MAF1 also enhances adipogenesis (*Chen et al., 2018*; *Figure 2—figure supplement 1*, *Figure 3—figure supplement 1*), we conclude that MAF1 is an important regulator in the development and differentiation of mesenchymal cells into multiple lineages.

MAF1 is a well-established repressor of RNA pol III-mediated transcription through its direct interaction with RNA pol III (*Vorländer et al., 2019*; *Vannini et al., 2010*). To further determine whether MAF1 functions to promote osteoblast differentiation through its ability to regulate RNA pol III-dependent transcription, we used complementary approaches to repress this transcription process. During differentiation, we observed an overall increase in tRNA gene transcripts. This increase was repressed by MAF1 overexpression, *Brf1* downregulation, or chemical inhibition of RNA pol III. Surprisingly, however, in contrast to the positive regulation of osteoblast differentiation by MAF1, chemical inhibition of RNA pol III or *Brf1* knockdown

**Table 1.** Summary of results found by distinct manipulations of RNA pol III-mediated transcription.

| Outcome/Phenotype | Mouse line Maf1-/- | Mouse line Prx1-Cre-MAF | ST2 cell line MAF1 OE | ST2 cell line shMAF1 | ST2 cell line shBrf1 | ST2 cell line ML-60218 |
|---|---|---|---|---|---|---|
| RNA pol III transcription | Increased | Decreased | Decreased | Increased | Decreased | Decreased |
| Bone mass | Increased | Increased | N/A | N/A | N/A | N/A |
| In vitro osteoblast differentiation/mineralization | Decreased | Increased | Increased | Decreased | Decreased | Decreased |
| In vivo bone marrow adipocyte number | Decreased | ND | N/A | N/A | N/A | N/A |
| In vitro adipocyte differentiation | Decreased | ND | Increased | ND | Increased | Increased |

resulted in a decrease in osteoblast differentiation (*Table 1*). Thus, while different perturbations in RNA pol III-dependent transcription all affect the differentiation process, MAF1-mediated changes produce an opposing action compared with chemical inhibition of RNA pol III or decreased *Brf1* expression. This is in contrast to what we observed for adipogenesis, where all three approaches to repress RNA pol III transcription similarly increased adipocyte formation (*Figure 2—figure supplement 1*, *Figure 3—figure supplement 1*, *Figure 4—figure supplement 2*, *Figure 5—figure supplement 1* and *Table 1*; *Chen et al., 2018*).

To understand the basis of the different osteoblast differentiation outcomes, we examined changes in gene expression that resulted from the different perturbations of RNA pol III-dependent transcription. RNA seq revealed that alterations in RNA pol III-mediated transcription prior to, and during differentiation, result in a limited number of overlapping changes, with the four conditions largely producing distinct changes in gene expression profiles. Changes in several established regulators of osteoblast formation and function were identified that correlated with the differentiation outcomes. Overall, the different gene expression profiles likely contribute to the differences in osteoblast differentiation that we observe. However, the precise mechanism underlying the difference between MAF1-regulated changes compared with the alternate approaches to repress RNA pol III-mediated transcription remains unclear. Unlike RNA pol III and Brf1, MAF1 is not an essential RNA pol III transcription factor, and it functions to repress this transcription process by directly interacting with RNA pol III (*Vorländer et al., 2019*). Given that MAF1 is recruited to certain RNA pol II promoters (*Johnson et al., 2007*; *Palian et al., 2014*; *Khanna et al., 2014*; *Li et al., 2016*), it is conceivable that the ability of MAF1 to regulate both RNA pol III- and RNA pol II-transcribed genes contributes to its ability to drive osteoblast differentiation. This may also explain its differential effect on osteoblast differentiation when compared with the alternative approaches used to selectively repress RNA pol III-dependent transcription. However, RNA seq did not detect changes in genes previously reported to be regulated by MAF1, such as *Tbp* and *Fasn*. The same finding was obtained in an RNA-seq analysis in *Maf1-/-* liver (*Bonhoure et al., 2020*). Thus, the regulation of RNA pol II-dependent gene expression by MAF1 is likely context dependent. Additional studies suggest that the regulation of RNA pol II targets by MAF1 is limited (*Orioli et al., 2016*). Therefore, we cannot exclude a possible role for MAF1 regulation of RNA pol II transcription in the regulation of osteoblast differentiation. However, since other approaches used to modulate RNA pol III-mediated transcription also alter osteoblast differentiation, MAF1 functions, at least in part, to regulate osteoblastogenesis through its effect on RNA pol III.

In addition to different effects on gene expression by manipulation of MAF1 expression, RNA-seq analysis revealed distinct changes in gene expression exhibited by chemical inhibition of RNA pol III and *Brf1* knockdown, despite similar outcomes on osteoblast differentiation. These data indicate that manipulating RNA pol III transcription using different approaches results in disparate outcomes on gene expression. This could be due to differential changes in the tRNA population. The prevalence of specific tRNAs has been correlated with codon-biased translation in multiple tissues (*Dittmar et al., 2006*; *Gobet et al., 2020*; *Kutter et al., 2011*; *Rak et al., 2018*). Our analysis of mRNA changes during osteoblast differentiation showed that a significant codon bias emerges during this process. Thus, it is conceivable that during osteoblast differentiation specific changes in tRNAs are required to efficiently drive codon-biased translation of mRNAs needed for differentiation to proceed. Thus, this may be a mechanism in which RNA pol III transcription affects osteoblast differentiation. There are several other mechanisms that could contribute to the observed differences in gene regulation by RNA pol III-mediated transcription. RNA pol III transcribes a variety of untranslated RNAs and changes

in any of these RNA pol III-derived transcripts may potentially play a role. Additionally, the generation of tRNA fragments (*Schimmel, 2018*; *Su et al., 2020*), or the regulation of RNA pol II genes through the recruitment of RNA pol III to nearby SINE sites such as described for *CDKN1a* (*Lee et al., 2015*), could all potentially contribute to the differential expression of osteoblast genes when RNA pol III-mediated transcription is altered. Future work will be needed to identify the specific mechanisms by which MAF1 and RNA pol III-mediated transcription alter gene expression to regulate osteoblast differentiation.

In all, our results describe a novel role for MAF1 and RNA pol III in bone biology. Given that different approaches used to modulate RNA pol III-dependent transcription also affect osteoblast differentiation, albeit in an opposing direction from MAF1-mediated effects, the findings support the idea that MAF1 functions, at least in part, to regulate osteoblast development and bone mass through its ability to control RNA pol III-mediated transcription. Distinct qualitative or quantitative changes resulting from different perturbations in RNA pol III-dependent transcription may also play a role in developmental disorders. Interestingly, several different syndromes relating to mutations in RNA pol III subunits show very heterogeneous phenotypes. This includes *POLR3*-related hypomyelinating leukodystrophies (*Lata et al., 2021*; *Yeganeh and Hernandez, 2020*; *Thomas and Thomas, 2019*), Wiedemann-Rautenstrauch syndrome, a neonatal progeroid syndrome (*Wu et al., 2021b*; *Wambach et al., 2018*; *Paolacci et al., 2017*; *Beauregard-Lacroix et al., 2020*), and cerebellar hypoplasia with endosteal sclerosis (*Terhal et al., 2020*; *Ghoumid et al., 2017*). Additionally, *Brf1* mutations have been shown to be causative for cerebellofaciodental syndrome (*Borck et al., 2015*; *Jee et al., 2017*; *Honjo et al., 2021*; *Valenzuela et al., 2020*). Some patients with RNA pol III-related mutations, but not all, show bone-related phenotypes (*Borck et al., 2015*; *Jee et al., 2017*; *Honjo et al., 2021*; *Terhal et al., 2020*; *Ghoumid et al., 2017*). The considerable heterogeneity of these syndromes has been suggested to be related to differential changes in RNA pol III-dependent transcription (*Yeganeh and Hernandez, 2020*). Thus, some cells and tissues, including bone and its cells, may be more sensitive to disruption of RNA pol III-mediated transcription. Together, these collective findings and our current study indicate that the exquisite regulation of RNA pol III plays an essential role in a variety of biological and developmental processes.

# Materials and methods

## Key resources table

| Reagent type (species) or resource | Designation | Source or reference | Identifiers | Additional information |
|---|---|---|---|---|
| Strain, strain background (*Mus musculus*) | *Rosa26*-Lox-stop-lox-*MAF1*-HA; LSL-*MAF1* | This paper | | An engineered construct of *Rosa26*-Lox-stop-lox-*MAF1*-HA was injected into C57Bl6/J mice embryonic stem cells. chimeric mice were created by by blastocyst injection of homologous recombinant clones. |
| Strain, strain background (*M. musculus*) | *Maf1*⁻/⁻ | *Bonhoure et al., 2015* | | Mouse line maintained in Dr. I Willis lab. |
| Strain, strain background (*M. musculus*) | *Prrx1*^Cre | Jackson laboratory | Strain #:005584 | |
| Cell line (*M. musculus*) | ST2 | RIKEN cell bank | #RCB0224 | |
| Transfected construct (*M. musculus*) | Scramble shRNA | Addgene, Sheila Steward | #17,920 | Lentiviral construct to express shRNA |
| Transfected construct (*M. musculus*) | MAF1 shRNA#1 | Millipore sigma | TRCN0000125776 | Lentiviral construct to express shRNA |
| Transfected construct (*M. musculus*) | MAF1 shRNA#2 | Millipore sigma | TRCN0000125778 | Lentiviral construct to express shRNA |
| Transfected construct (*M. musculus*) | Brf1 shRNA#1 | Millipore sigma | TRCN0000119897 | Lentiviral construct to express shRNA |
| Transfected construct (*M. musculus*) | Brf1 shRNA#2 | Millipore sigma | TRCN0000119901 | Lentiviral construct to express shRNA |
| Transfected construct (*M. musculus*) | pInducer20 | Addgene Stephen Elledge | #44,012 | Lentiviral construct to express shRNA |

*Continued on next page*

*Continued*

| Reagent type (species) or resource | Designation | Source or reference | Identifiers | Additional information |
|---|---|---|---|---|
| Transfected construct (Human) | pInducer20-*MAF1*-HA | This paper | | pInd20-*MAF1*-HA was cloned by taking MAF1-HA from pFTREW-MAF1-HA into a pInducer20 construct by gateway cloning using LR clonase. Cell line *M. musculus* construct: human |
| Chemical compound, drug | Calcein | Millipore Sigma | C0875 | 10 mg/kg |
| Chemical compound, drug | Xylenol orange | Millipore Sigma | X0127 | 90 mg/kg |
| Chemical compound, drug | LR clonase | Thermo Fisher | #11791020 | |
| Chemical compound, drug | Doxycycline hyclate | Millipore Sigma | #D9891 | Used at 1 µM |
| Chemical compound, drug | ML-60218 | Millipore Sigma | #557,403 | RNA pol III inhibitor |
| Chemical compound, drug | Ascorbic acid | Sigma | #A4544 | Used at 50 µg/mL |
| Chemical compound, drug | Β-glycerolphosphate | Millipore Sigma | #35,675 | Used at 10 mM |
| Chemical compound, drug | Cetylpyridinium chloride | Sigma | #C0732 | Used at 10% for alizarin red extraction |
| Chemical compound, drug | rosiglitazone | Sigma | R2408 | Used at 1 µM |
| Chemical compound, drug | 3-isobutyl-1-methyl xanthine | Sigma | I5879 | Used at 0.5 mM |
| Chemical compound, drug | dexamethasone | Sigma | D4902 | Used at 2 µM |
| Chemical compound, drug | Insulin | Sigma | I05016 | Used at 10 µg/mL |
| Chemical compound, drug | RNA stat-60 | Tel-test Inc | #NC9256697 | |
| Chemical compound, drug | Alizarin Red | Sigma | #A5533 | Used at 1% at ph 4.2 |
| Chemical compound, drug | Oil red O | Sigma | #01391 | Used at 0.3% |
| Chemical compound, drug | collagenase IV | Gibco | #17104019 | Used at 2.5% |
| Commercial assay or kit | TRAP staining kit | Sigma | #387A-1KT | |
| Commercial assay or kit | Von Kossa staining | Statlab | #KTVKO | |
| Commercial assay or kit | Alkaline phosphatase staining | Vector laboratories | #SK5300 | |
| Commercial assay or kit | Quick-RNA miniprep kit | Zymo | #R1055 | Used for RNA isolation from cell culture |
| Commercial assay or kit | Direct-zol RNA miniprep kit | Zymo | #R2052 | Used for RNA isolation from femurs |
| Commercial assay or kit | Superscript IV First Strand Synthesis Kit | Invitrogen | #18091050 | cDNA synthesis |
| Commercial assay or kit | SYBR fast qPCR mastermix | KAPA Biosystems | #KK4602 | |
| Peptide, recombinant protein | M-CSF | Peprotech | #300–25 | Used at 30 ng/mL |
| Peptide, recombinant protein | RANK-L | Peprotech | #310–01 C | Used at 100 ng/mL |
| Peptide, recombinant protein | FGF2 | Biovision | #4,038 | Used at 10 ng/mL |

*Continued on next page*

*Continued*

| Reagent type (species) or resource | Designation | Source or reference | Identifiers | Additional information |
|---|---|---|---|---|
| Commercial assay or kit | DC protein assay | Biorad | #5000112 | |
| Antibody | Anti-MAF1 (H2) (mouse monoclonal) | Santa Cruz | #SC-515614 | (Wb 1:500) |
| Antibody | Anti-TFIIIB90 (mouse monoclonal) | Santa Cruz | #SC-390821 | Antibody to Brf1. (Wb 1:1000) |
| Antibody | Anti-VINCULIN (mouse monoclonal) | Santa Cruz | # sc-73614 AF488 | (Wb 1:5000) |
| Antibody | Anti-RUNX2 (rabbit monoclonal) | Cell Signaling | #12,556 | (Wb 1:1000) |
| Antibody | Anti-PPARγ (rabbit monoclonal) | Cell Signaling | #2,435 | (Wb 1:1000) |
| Antibody | Anti-FABP4 (rabbit monoclonal) | Cell Signaling | #3,544 | (Wb 1:1000) |
| Antibody | Anti-HA (Rat monoclonal) | Roche | #11867423001 | (Wb 1:1000) |
| Software, algorithm | R- studio | https://rstudio.com | | Version 4.1.1 |
| Software, algorithm | DeSeq2 | 10.18129/B9.bioc.DESeq2 | | |
| Software, algorithm | clusterProfiler | doi.org/10.1016 /j.xinn.2021.100141 | | |
| Software, algorithm | InteractiVenn | 10.1186 /s12859-015-0611-3 | | |
| Software, algorithm | Graphpad prism | https://www.graphpad.com/ | | Version 9.3.1 |

## Mouse lines and bone analyses

All mouse experiments were performed according to a protocol approved by the Institutional Animal Care and Use Committee at Baylor College of Medicine and Albert Einstein College of Medicine. *Rosa26*-Lox-stop-lox-MAF1-HA (LSL-MAF1) mice were generated by injecting an engineered construct into mouse (C57Bl6/J strain) embryonic stem cells and selecting for homologous recombinant clones. The selected clones were used to generate chimeric mice by blastocyst injection. The chimeric mice were bred to found the LSL-MAF1 colony. *Prx1*-Cre lines were a kind gift from Dr. Brendan Lee (Baylor College of Medicine). LSL-MAF1 mice were mated with *Prx1*-Cre for conditional overexpression of MAF1-HA. Littermate controls expressing only Cre were used as a control. To measure dynamic bone histomorphometric parameters, mice were injected with calcein (Sigma) 10 mg/kg at a 6-day interval, 8 and 2 days before euthanasia. Left femurs were collected for µCT histomorphometry at 12 weeks. They were fixed for 48 hr in 4% paraformaldehyde (PFA) and stored at 4 °C in 70% ethanol. µCT of left femurs was performed using the Scanco µCT-40 system at 16 µm resolution. About 75 slices in the metaphyseal region of each femur were analyzed, starting at 10 slices beyond disappearance of the growth plate. Histomorphometry measurements were performed by the bone histomorphometry core at M.D. Anderson Cancer Center (Houston, TX). Tibiae and right femurs were dissected, bone marrow was washed out, and bones were subsequently snap frozen in liquid nitrogen for subsequent protein and RNA isolation. For the *Maf1*[-/-] mice, 12-week-old mice were used for µCT and histomorphometry. Mice were double labeled by injecting Calcein (Sigma) 10 mg/kg 8 days before sacrifice and Xylenol Orange (Sigma) at 90 mg/kg 2 days before euthanasia. We calculated traditional metrics for bone formation through manual imaging and morphometry on blinded samples. Derived parameters include mineralized surfaces (MS), mineral apposition rate (MAR), and bone formation rate (BFR). µCT measurements at the spine, femurs and tibiae were performed through the courtesy of Dr. Jay Cao (USDA, North Dakota) using a Scanco µCT-40 scanner.

## ST2 cell culture and differentiation

ST2 cells were acquired from RIKEN BRC cell bank. Cells, which tested negative for mycoplasma, were grown in basic medium, ascorbic acid free α-MEM (Caisson laboratories) supplemented with 10%

FBS (Gibco). For the knockdown experiments, cells were infected with a scrambled control gift from Sheila Stewart addgene #17,920 (*Saharia et al., 2008*), *MAF1* shRNA (#1 TRCN0000125776 and #2 TRCN0000125778), or *Brf1* shRNA (#1 TRCN0000119897 or #2 TRCN0000119901). pInd20-MAF1-HA was cloned by taking MAF1-HA from pFTREW-MAF1-HA (*Palian et al., 2014*) into a pInducer20 construct by gateway cloning using LR clonase (Thermo Fisher). The empty pInducer20 vector was a gift from Stephen Elledge (Addgene #44012) (*Meerbrey et al., 2011*). Virus production and cell infection was performed as described previously (*Chen et al., 2018*). Cells were used for differentiation within three passages of selection. For MAF1 overexpression, pInducer20-MAF1HA infected, or pInducer20-empty cells were treated with 1 μM Dox 24 hr before differentiation was started. For ML-60218 treatment, cells were treated starting 24 hr before differentiation with 40 μM ML-60218 in DMSO (Millipore) or an equal volume of DMSO as control. ML-60218 treatment continued for 2 days after differentiation was initiated after which the compound was removed. For osteoblast differentiation, ST2 cells were plated at $1.8 \times 10^5$ cells *per* well in a 6-well plate. Cells were grown to confluence after which osteoblast differentiation medium, basic media with 50 μg/mL ascorbic acid and 10 mM β-glycerolphosphate was added (day 0) and changed every two days. For adipocyte differentiation, ST2 cells were plated at $1.8 \times 10^5$ cells *per* well in a 6-well plate and grown to confluency. On day 0, adipogenic medium was added (basic media with 1 μM rosiglitazone, 0.5 mM 3-isobutyl-1-methyl xanthine, 2 μM dexamethasone, and 10 μg/mL insulin). After 2 days, the media was changed to maintenance medium (basic media with 10 μg/mL insulin), which was changed every 2 days for the remainder of the experiment. For in vitro experiments, each experiment was performed using three independent replicates and repeated at least three times. One representative experiment is shown.

## Osteoclast cultures

Bone marrow cells were isolated from femora and tibiae of *Maf1*$^{-/-}$ and WT mice in alpha-MEM. Cells were cultured for 2 days with M-CSF (30 ng/mL). Non-adherent cells were collected and purified by Ficoll-Plus (Amersham Pharmacia). They were then incubated with M-CSF (30 ng/mL) and RANK-L (100 ng/mL) for 4–6 days followed by staining for Tartrate-resistant acid phosphatase (TRAP) using a kit (Sigma) *per* manufacturer's instruction. The number of TRAP-positive cells was counted.

## Cfu-f and Cfu-ob cultures

Marrow stromal cells were cultured in the presence of ascorbate-2-phosphate (1 mM) (Sigma). Colony-forming units-fibroblastoid (Cfu-f) and colony-forming units-osteoblastoid (Cfu-ob) were counted, respectively, following alkaline phosphatase staining after 14 day cultures, or von Kossa staining after 21 day cultures.

## Primary stromal cell culture

Primary stromal cells were isolated from 6- to 8-week-old Prx1-Cre; LSL-MAF1-HA mouse femurs and tibiae. Bones were dissected, cleaned and the marrow was flushed out. Bone pieces were digested using 2.5% collagenase IV (Gibco) for 2–4 hr at 37°C. Cells were strained and maintained in basic medium with 10 ng/mL FGF2 (Biovision). For differentiation, cells were plated in a 48 or 6 well plates and differentiation was performed as described above.

## RNA isolation and quantitative PCR

Total RNA from cells was isolated using the quick-RNA miniprep kit (Zymo Research) following manufacturer's protocol. For femurs, samples were ground using mortar and pestle in liquid nitrogen, and then further disrupted in RNA stat-60 (Tel-Test Inc) using a polytron. RNA was isolated using the Direct-zol miniprep kit (Zymo Research). cDNA was synthesized using Superscript IV First Strand Synthesis Kit (Invitrogen). Quantitative PCR was performed using SYBR fast qPCR mastermix (KAPA Biosystems) on the Roche 480 Lightcycler. Gene-specific primers are described in *Supplementary file 1*. RNA was quantified relative to *Ef1a* for osteoblast differentiation and *Ppia1* for adipocyte differentiation unless otherwise denoted.

## Protein isolation

Cells were washed twice and lysed in RIPA buffer. Tibia were ground in mortal and pestle, and further disrupted in RIPA buffer using a polytron. Samples were then sonicated. Cell lysate concentrations

were measured using DC protein assay (Biorad) and similar amounts of protein lysate were loaded. The following antibodies were used: MAF1 (H2), TFIIIB90 (Brf1) (A8), Vinculin (7F9), and β-actin (C4) (Santa Cruz), Runx2, Ppary, Fabp4 (Cell signaling) and HA (Roche).

## Staining

Cells were fixed for 10 min in 4% PFA, washed twice with PBS and once with water. Oil red O staining was performed using 0.3% Oil Red O solution (Sigma). Alkaline phosphatase staining used an alkaline phosphatase blue substrate kit (Vector Laboratories). Alizarin red staining was performed using 1% alizarin red (Sigma-Aldrich) at a pH of 4.2. Cell counts were taken using the Cytation 5 Microscope. Alizarin Red was extracted using 10% cetylpyridinium chloride (CPC) and absorption was measured at 563 nM.

## Sequencing

ST2 cells were prepared, plated, and differentiated as described above. shBrf1#2 and shMAF1#2 were used for sequencing analysis. Triplicates of each sample were used for each analysis and RNA was extracted on day 0 and day 4. For each replicate, three wells of a 6-well plate were combined. For each condition, triplicate RNA was isolated using the quick RNA miniprep kit (Zymo Research). Library preparation and RNA-seq were performed by Novogene Co. (Sacramento, CA, USA). Differentially expressed genes were determined using DESeq2 with FDR <0.05 and |log2foldchange| >0.7. One replicate of sh*Brf1* at day 0 was considered an outlier by principal component analysis and hierarchical clustering and removed from analysis. GO analysis was performed using the clusterProfiler R package (*Wu et al., 2021a*). Venn diagrams were made using InteractiVenn (*Heberle et al., 2015*). For codon usage analysis, an exhaustive list of gene-coding sequences was obtained from GENCODE (M27) and codon use rates were calculated. For each codon, a selection rate against other potential isodecoders was determined. Gene subsets were established from alteration in RNAseq at over log2foldchange 0.7 in either direction and padj<0.05 by DESeq2, or by membership in gene ontology as osteoblast differentiation. For comparisons between two groups two-tailed Student's t-test were performed followed by Benjamini-Hochberg correction of the full comparison set.

## Statistical analysis

For comparisons between two groups, two-tailed Student's t-test were performed. For comparisons with more than two groups, ANOVA was used followed by paired t-tests with Holm correction. Significance was determined at p<0.05.

## Acknowledgements

We would like to thank the Genetically Engineered Rodent Models Core at the Baylor College of Medicine for assistance with mouse model production. Resources accessed through the core were supported by a National Institutes of Health grant (P30CA125123) to the Dan L Duncan Comprehensive Cancer Center. We would like to thank Brian Dawson at Baylor College of Medicine for his help with the μCT measurements. This work was supported by NIH grants R01 CA108614 and R01 CA74138 (to DLJ), R01 GM120358 (to IMW), NIH--U19 AG060917 (MZ), R01 AG071870, U01 AG073148, and R01AG074092 (to MZ and TY). MZ also thank the Harrington Discovery Institute for the Innovator–Scholar Award.

## Additional information

### Competing interests

Mone Zaidi: Senior editor, *eLife*. The other authors declare that no competing interests exist.

## Funding

| Funder | Grant reference number | Author |
|--------|------------------------|--------|
| National Cancer Institute | CA108614 | Tony Yuen<br>Ian M Willis<br>Mone Zaidi<br>Clifford J Rosen |
| National Cancer Institute | CA74138 | Li Sun |
| National Institutes of Health | | Deborah L Johnson<br>Mone Zaidi<br>Tony Yuen |
| Baylor College of Medicine | | Deborah L Johnson |
| Discovery Institute | | Mone Zaidi |

The funders had no role in study design, data collection and interpretation, or the decision to submit the work for publication.

## Author contributions

Ellen Phillips, Conceptualization, Data curation, Formal analysis, Investigation, Methodology, Validation, Visualization, Writing – original draft, Writing – review and editing; Naseer Ahmad, Li Sun, Data curation, Investigation, Methodology; James Iben, Formal analysis, Software, Validation, Writing – review and editing; Christopher J Walkey, Conceptualization, Investigation, Resources; Aleksandra Rusin, Methodology, Supervision; Tony Yuen, Validation, Writing – review and editing; Clifford J Rosen, Data curation, Formal analysis, Investigation; Ian M Willis, Project administration, Resources, Supervision, Writing – review and editing; Mone Zaidi, Investigation, Methodology, Supervision, Writing – review and editing; Deborah L Johnson, Conceptualization, Formal analysis, Funding acquisition, Methodology, Project administration, Resources, Supervision, Writing – original draft, Writing – review and editing

## Author ORCIDs

Ellen Phillips (iD) http://orcid.org/0000-0003-3654-2066
Naseer Ahmad (iD) http://orcid.org/0000-0001-9514-2703
Ian M Willis (iD) http://orcid.org/0000-0001-6599-2395
Mone Zaidi (iD) http://orcid.org/0000-0001-5911-9522
Deborah L Johnson (iD) http://orcid.org/0000-0003-1745-1580

## Ethics

This study was performed in strict accordance with the recommendations in the Guide for the Care and Use of Laboratory Animals of the National Institutes of Health. All of the mice were handled according to approved institutional animal care and use committee (IACUC) protocol AN-6370 of Baylor College of Medicine.

## Decision letter and Author response

Decision letter https://doi.org/10.7554/eLife.74740.sa1
Author response https://doi.org/10.7554/eLife.74740.sa2

# Additional files

## Supplementary files

- Supplementary file 1. Maf1-/- mice show increased bone mass in the spine.
- Transparent reporting form

## Data availability

Sequencing data will be deposited in GEO Source data files will be provided All data will be made available to the public. Raw and processed data from the RNA sequencing experiment determining gene expression before and during osteoblast differentiation has been uploaded to the GEO data base with accession nr. GSE203308.

The following dataset was generated:

| Author(s) | Year | Dataset title | Dataset URL | Database and Identifier |
|---|---|---|---|---|
| Phillips E, Ahmad N, Sun L, Iben J, Walkey CJ, Rusin A, Yuen T, Rosen CJ, Willis IM, Zaidi M, Johnson DL | 2022 | RNA seq after RNA pol III manipulation by Brf1 and Maf1 knockdown, Maf1 overexpression and ML60218 treatment in ST2 cells before, and during differentiation into osteoblasts | https://www.ncbi.nlm.nih.gov/geo/query/acc.cgi?acc=GSE203308 | NCBI Gene Expression Omnibus, GSE203308 |

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

## Appendix 1

**Appendix 1—table 1.** qPCR primers used for genotyping and qRT-PCR analysis.

| Target | Forward primer | Reverse primer | citation |
|---|---|---|---|
| Cre (genotyping) | TCCAATTTACTGAC CGTACACCAA | CCTGATCCTGGC AATTTCGGCTA | |
| LSL-MAF1 (genotyping) | TTCACTTCATAC CCATACGACG | CCATTTTCCTTA TTTGCCCCTA | |
| WT Maf1 | AGGCTTGCAGG GCAGCAATG | CACTGGCTGACA GGGAGATG | Bonhoure et al., 2015 |
| Maf1 KO (genotyping) | AGGCTTGCAGG GCAGCAATG | TGGCCCTTAGAG CTGGAGTG | Bonhoure et al., 2015 |
| Pre-tRNA$^{Leu}$ | GTCAGGATGGCC GAGTGGTCTAAG | CCACGCCTCCATACGGA GAACCAGAAGACCC | Chen et al., 2018 |
| Pre-tRNA$_i^{Met}$ | CTGGGCCCAT AACCCAGAG | TGGTAGCAGA GGATGGTTTC | Chen et al., 2018 |
| Pre-tRNA$^{Ile}$ | GTTAGCGCGC GGTACTTATA | GGATCGAACT CACAACCTCG | Graczyk et al., 2015 |
| Pre-tRNA$^{Pro}$ | GGCTCGTTGGTCTAGGG | TTTGAACCCGGGACCTC | Graczyk et al., 2018 |
| Maf1 | GACTATGACTTC AGCACAGCC | CTGGGTTATAGC TGTAGATGTCAC | Chen et al., 2018 |
| Brf1 | GGAAAGGAATCAAG AGCACAGACCC | GTCCTCGGGTAA GATGCTTGCTT | Chen et al., 2018 |
| Runx2 | AGGGACTATGG CGTCAAACA | GGCTCACGT CGCTCATCTT | Fujioka-Kobayashi et al., 2016 |
| Col1a1 | CCCAATGGTG AGACGTGGAA | TTGGGTCCCT CGACTCCTAC | |
| Sp7 | ATGGCGTCCT CTCTGCTTG | GTCCATTGGT GCTTGAGAAGG | Fitter et al., 2017 |
| Bglap | TCTGACAAAG CCTTCATGTCC | AAATAGTGATA CCGTAGATGCG | Pustylnik et al., 2013 |
| Alp | CGGATCCTGA CCAAAAACC | TCATGATGT CCGTGGTCAAT | |
| Ibsp | GAAAATGGAG ACGGCGATAG | CATTGTTTTC CTCTTCGTTTGA | |
| EF1a | CTGAACCATC CAGGCCAAAT | GGCTGTGT GACAATCCAG | Van Itallie et al., 2006 |
| β-actin | CGACAACGGC TCCGGCATG | CTGGGGTGTTGAA GGTCTCAAACATG | |
| Rankl | CAGCCATTTGC ACACCTCAC | GTCTGTAGGT ACGCTTCCCG | |
| Opg | AGGAACTGCA GTCCGTGAAG | ATTCCACACT TTTGCGTGGC | |
| Ppia1 | CGAGCTGTTTGCAG ACAAAGTTCC | CCCTGGCACA TGAATCCTGG | Chen et al., 2018 |
| Pparg | ATCATCTACACG ATGCTGGCCT | TGAGGAACTCC CTGGTCATGAATC | Chen et al., 2018 |
| Pparg2 | TCGCTGATGCA CTGCCTATGA | GGAGAGGTC CACAGAGCTGAT | |
| Cebpa | GAACAGCAACGA GTACCGGGTA | CCATGGCCTT GACCAAGGAG | Chen et al., 2018 |

*Appendix 1—table 1 Continued on next page*

*Appendix 1—table 1 Continued*

| Target | Forward primer | Reverse primer | citation |
|--------|----------------|----------------|----------|
| *Fabp4* | TGGGAACCTG GAAGCTTGTCT | TCGAATTCCAC GCCCAGTTTGA | *Chen et al., 2018* |

