## [Editor Report]

In this manuscript, Phillips et al., have used several complementary in vivo and in vitro approaches to analyze the effects of regulated MAF1 expression or inhibition of RNA pol III transcription on osteogenesis and adipocyte differentiation. The data are well controlled and of excellent quality, providing novel insights into Maf1 and RNA polymerase-mediated transcriptions in skeleton biology.

---

## [Decision Letter]

**Decision letter after peer review:**

Thank you for submitting your article "Maf1, a repressor of RNA polymerase III-dependent transcription, regulates bone mass" for consideration by *eLife*. Your article has been reviewed by 3 peer reviewers, and the evaluation has been overseen by a Reviewing Editor and Carlos Isales as the Senior Editor. The reviewers have opted to remain anonymous.

All three reviewers and the Reviewing Editor agree that the paper is novel, well written and the findings in general support the conclusions. However, one major concern raised by the reviewers relates to the conflicting data between the in vivo and in vitro models. Other issues relate to not testing the prediction that codon bios occurs when RNA Pol III is altered and the failure to address the phenotypic differences caused by lack of Maf1 between the published and the current study. Please see the recommendations for authors below.

*Reviewer #1 (Recommendations for the authors):*

The manuscript is clearly written, and the figures are well presented. Most in vivo and in vitro data were sound and properly analyzed, therefore, supported the conclusions.

The authors should perform bone histology and histomorphometric analyses of the number and activities of osteoblasts and osteoclasts in vivo in vertebra or tibia of Maf1-/- and Prx-Cre;LSL-Maf1 mice to uncover the in vivo mechanisms of high bone mass in both strains of mice.

*Reviewer #2 (Recommendations for the authors):*

The authors should keep in mind that the conflicting results obtained in mice, derived cells, and ST2 cells could be due to differences resulting from 2D or 3D growth of the cells. ST2 cells were subjected to 2D growth only, stromal cells were subjected to 3D and 2D growth, and the mice were subjected exclusively to 3D growth. This could have a strong impact on the functions of MAF1. This aspect could be discussed.

The authors should also analyze the expression of the U6 gene or other type 3 promoter-driven genes. ML-60218, BRF1 KD, or changes in MAF1 expression will affect the expression of these genes in different ways. Their impact on bone differentiation should not be neglected.

In the second paragraph on page 5, the sentence "The interventions used to manipulate RNA pol III-dependent transcription resulted in significant changes in gene expression at both day 0 and day 4" contains a superfluous "of."

*Reviewer #3 (Recommendations for the authors):*

1. The idea of codon bias, which when combined with the pool of available tRNA that are regulated by RNApolIII is VERY interesting. The authors state, "This suggests, for the first time, that codon bias may play a role during osteoblast differentiation", however, the authors should put this idea to test in the manipulations done in this paper. What happens to this codon bias when RNA pol III is changed? At a minimum please do the analysis relative to the differences in tRNAs being made in cells with Maf1 oe versus Brf inhibition?

2. The data strongly supports the notion that Maf1 promotes osteoblast differentiation and mineralization. However, Willis and colleagues previously demonstrated that the Maf1KO mouse is long-lived. In the absence of Maf1 of bone density would be impaired and bone biology (homeostasis) would be disrupted, which would predict a frail state. The authors need to better reconcile their observations with the previous literature suggesting that loss of Maf1 is a good thing at the organism-level.

3. Related to #2 above, the authors should include in their discussion the scientific basis for the notion of cell non-autonomous effects on bone length, which oppose the ex vivo data. Based on their data and the work in the field are there candidates molecules or pathways that could support this idea? or perhaps even we know what it is not? This would also clarify the inclusion of the adipocyte data, which unlike the osteoblast data, Maf1 plays a consistent in the in vivo and ex vivo models. The difference in response in vivo and the ex vivo models is intriguing, but in its current form is only descriptive and does not provide a mechanistic basis.

4. Are the effects of Maf1 overexpression dependent upon the inhibition of RNApolIII or are there independent effects of Maf1oe that could be at play here? Perhaps this could explain why the in vivo and ex vivo models are different for osteoblasts differentiation but are aligned for adipocytes? The authors have very nice tools to directly test this idea, which may also shed light on the issues raised above.

---

## [Author Response]

Reviewer #1 (Recommendations for the authors):The manuscript is clearly written, and the figures are well presented. Most in vivo and in vitro data were sound and properly analyzed, therefore, supported the conclusions.The authors should perform bone histology and histomorphometric analyses of the number and activities of osteoblasts and osteoclasts in vivo in vertebra or tibia of Maf1-/- and Prx-Cre;LSL-Maf1 mice to uncover the in vivo mechanisms of high bone mass in both strains of mice.

We performed histomorphometric analysis and added the data in Supplemental Figures 2 and 4. The data indicates that the increased bone mass observed in *Maf1^-/-^* mice is due to increased bone formation. However, the signal(s) that serve to increase bone formation likely originates from a different tissue other than the bone since the ex vivo data shows that when the cells are isolated from external signals, less osteoblast formation and increased osteoclast formation occurs.

The histomorphometry analysis performed on the Prx1-Cre-MAF1-HA mice confirms that these mice display an increase in bone mass in the femur. However, osteoblast and osteoclast numbers and dynamic histomorphometry does not show statistically significant changes. Thus, we cannot draw definitive conclusions from this data. However, there does appear to be a trend towards increased bone formation and increased osteoblast numbers. This, in combination with our other results, suggests that the increase in bone mass in these mice is due to increased osteoblast differentiation or function.

Reviewer #2 (Recommendations for the authors):The authors should keep in mind that the conflicting results obtained in mice, derived cells, and ST2 cells could be due to differences resulting from 2D or 3D growth of the cells. ST2 cells were subjected to 2D growth only, stromal cells were subjected to 3D and 2D growth, and the mice were subjected exclusively to 3D growth. This could have a strong impact on the functions of MAF1. This aspect could be discussed.

We have amended the discussion to acknowledge the possibility that different culture methods may affect our conclusions. However, all our in vitro data are consistent with results demonstrating that Maf1 enhances osteoblast differentiation. Both Maf1 knockdown in ST2 cells and Maf1 deficiency in primary stromal cells results in decreased osteoblast differentiation, while Maf1 overexpression in ST2 cells and primary stromal cells enhances osteoblast differentiation. Additionally, Maf1 overexpression in the Prx1-Cre-MAF1-HA mouse model increased bone mass. The only result that was inconsistent with these data is the increase in bone mass observed in the whole body Maf1 knock out. However, since this model represents a whole-body deficiency in Maf1, this suggests that the loss of Maf1 in other tissues, outside of the bone, ultimately affects the bone phenotype. Different effects based on time or tissue specific expression changes is not uncommon in bone studies (Liu et al., 2016).

The authors should also analyze the expression of the U6 gene or other type 3 promoter-driven genes. ML-60218, BRF1 KD, or changes in MAF1 expression will affect the expression of these genes in different ways. Their impact on bone differentiation should not be neglected.

As discussed above, it is conceivable that the expression of a variety of RNA pol III-derived transcripts impact osteoblast differentiation. While Maf1 (Chen et al., 2018) and ML-60218 (Pagano et al., 2007) have been shown to repress the transcription from all type 1, 2and 3 RNA pol III-dependent promoters, Brf1 knockdown only represses type 1 and 2 promoters. Thus, Brf1 knockdown would not affect U6 or other type 3-driven promoters. Therefore, it is unlikely that changes in the expression of type 3 promoters account for the distinct effect of Maf1 on osteoblast differentiation compared to Brf1 knockdown and ML-60218 treatment. Our analysis focused on the resultant changes on pre-tRNA synthesis to ensure that the conditions used (ML60218 treatment, Brf1 knockdown, and Maf1 expression changes) were affecting RNA pol IIImediated transcription.

Although we are unable to determine possible changes in all the different RNA pol III-derived transcripts, it is conceivable that multiple targets play a role. The discussion was edited to acknowledge the possibility that in addition to specific changes in tRNAs, other RNA pol IIIderived RNAs could also regulate osteoblast differentiation.

In the second paragraph on page 5, the sentence "The interventions used to manipulate RNA pol III-dependent transcription resulted in significant changes in gene expression at both day 0 and day 4" contains a superfluous "of."

We have amended this accordingly.

Reviewer #3 (Recommendations for the authors):1. The idea of codon bias, which when combined with the pool of available tRNA that are regulated by RNApolIII is VERY interesting. The authors state, "This suggests, for the first time, that codon bias may play a role during osteoblast differentiation", however, the authors should put this idea to test in the manipulations done in this paper. What happens to this codon bias when RNA pol III is changed? At a minimum please do the analysis relative to the differences in tRNAs being made in cells with Maf1 oe versus Brf inhibition?

We agree that this would be a very important experiment, Therefore, we enlisted Array Star to perform tRNA sequencing analysis. They analyzed the tRNA population during osteoblast differentiation, and for all the conditions where RNA pol III transcription was manipulated. Unfortunately, we were unable to derive useful information from these analyses. The experiment showed a high signal to noise ratio that obscured our ability to draw conclusions. tRNA sequencing technology currently has several caveats. The highly modified nature of tRNAs poses a challenge in preparing a library, and the highly repetitive nature of tRNAs makes it difficult to accurately align and quantify tRNAs (Behrens et al., 2021). Because of this, even with the highest standards currently available, the technique remains error prone.

When we examined changes in tRNAs during differentiation, we found that differentiation affects the tRNA pool. However, only a very small number of tRNAs reached statistical significance in our analysis. Additionally, while some tRNAs trended to change in a similar direction as the codon bias analysis during differentiation, we were not able to identify an overall significant correlation that would allow us to make definitive conclusions.

Comparing how the different approaches to manipulate RNA pol III affected the tRNA population, only a few tRNAs showed significant changes. It has previously been shown that changes in a single tRNA can have a biological phenotype (Goodarzi et al., 2016). Therefore, these changes could be biologically relevant. However, our analysis does not currently have the statistical power to draw this conclusion. The tRNA populations were different between the groups, suggesting that different manners of manipulating RNA pol III have a diverse effect on tRNA expression. Since only a few tRNAs reached statistical significance, it is possible that this is an artifact of the overall low significance of the analysis. We believe there are likely more tRNAs changes that we were not able to detect due to the current technical limitations. Therefore, while we agree that the codon bias question is very interesting and worth pursuing, it proved to be challenging within experimental (and time) constraints. While we feel there is may be something there, we didn’t have the statistical power to provide any definitive conclusions.

2. The data strongly supports the notion that Maf1 promotes osteoblast differentiation and mineralization. However, Willis and colleagues previously demonstrated that the Maf1KO mouse is long-lived. In the absence of Maf1 of bone density would be impaired and bone biology (homeostasis) would be disrupted, which would predict a frail state. The authors need to better reconcile their observations with the previous literature suggesting that loss of Maf1 is a good thing at the organism-level.

While decreasing Maf1 results in decreased osteoblast differentiation and function, the Maf1^/-^ mouse overall has denser bones. This is likely due to signals originating in other tissues due to Maf1 loss. Thus, we do not see a reduction in bone density in these mice corresponding to longevity. Future studies will be needed to determine the role of Maf1 in other tissues leading to increased bone density and other biological effects.

3. Related to #2 above, the authors should include in their discussion the scientific basis for the notion of cell non-autonomous effects on bone length, which oppose the ex vivo data. Based on their data and the work in the field are there candidates molecules or pathways that could support this idea? or perhaps even we know what it is not? This would also clarify the inclusion of the adipocyte data, which unlike the osteoblast data, Maf1 plays a consistent in the in vivo and ex vivo models. The difference in response in vivo and the ex vivo models is intriguing, but in its current form is only descriptive and does not provide a mechanistic basis.

Several studies have shown distinct effects on bone density depending on the time and the tissues in which they are expressed. An example of this is Notch (Liu et al., 2016). There are many potential candidate pathways and signals that originate from other tissues to affect bone density. Examples of these include but are not limited to. Vitamin D (Goltzman, 2018), oxytocin (Breuil et al., 2021), parathyroid hormone, (Wein and Kronenberg, 2018) leptin, (Reid et al., 2018) and adiponectin (Naot et al., 2016). Each of these originate from several different tissues. While this includes the possibility of signaling that may occur from adipocytes, we cannot exclude a multitude of other tissues that might play a role and we cannot determine the extent of the role of adipocytes. We do not currently know the role of Maf1 in the regulation of these various components. While there is a wide variety of candidates, tissue specific Maf1 knockout in a variety of tissues would be needed to determine what tissues/cells affect bone density and the signals and mechanisms that play a role. This is currently beyond the scope of this paper.

4. Are the effects of Maf1 overexpression dependent upon the inhibition of RNApolIII or are there independent effects of Maf1oe that could be at play here? Perhaps this could explain why the in vivo and ex vivo models are different for osteoblasts differentiation but are aligned for adipocytes? The authors have very nice tools to directly test this idea, which may also shed light on the issues raised above.

Several RNA pol II transcribed genes have been shown to be regulated by Maf1. Therefore, we cannot exclude the possibility that Maf1 may have additional effects beyond its function as a repressor of RNA pol III, including directly or indirectly regulating RNA pol II genes, to affect osteoblast differentiation. However, because the mechanism by which Maf1 regulates RNA pol II is currently unknown, we cannot separate its effects on repressing transcription from RNA pol III and RNA pol II genes. It has previously been reported that the vast majority of Maf1 targets consists of RNA pol III sites (Orioli et al., 2016). Additionally, in our RNA seq analysis, we saw no effect of Maf1 manipulation on previously reported RNA pol II targets TBP, Fasn, and Pten. Thus, while Maf1 may play additional, perhaps yet to be determined roles, we hypothesize that its main function occurs through its ability to repress RNA pol III-mediated transcription.

References

Behrens, A., Rodschinka, G., and Nedialkova, D. D. (2021). High-resolution quantitative profiling of tRNA abundance and modification status in eukaryotes by mim-tRNAseq. Molecular Cell, 81(8), 1802-1815.e7. https://doi.org/10.1016/J.MOLCEL.2021.01.028

Bonhoure, N., Byrnes, A., Moir, R. D., Hodroj, W., Preitner, F., Praz, V., Marcelin, G., Chua, S. C., Martinez-Lopez, N., Singh, R., Moullan, N., Auwerx, J., Willemin, G., Shah, H., Hartil, K., Vaitheesvaran, B., Kurland, I., Hernandez, N., and Willis, I. M. (2015). Loss of the RNA polymerase III repressor MAF1 confers obesity resistance. Genes and Development, 29(9),

934–947. https://doi.org/10.1101/gad.258350.115

Bonhoure, N., Praz, V., Moir, R. D., Willemin, G., Mange, F., Moret, C., Willis, I. M., and Hernandez, N. (2020). MAF1 is a chronic repressor of RNA polymerase III transcription in the mouse. Scientific Reports, 10(1), 11956. https://doi.org/10.1038/s41598-020-68665-0

Breuil, V., Trojani, M.-C., and Ez-Zoubir, A. (2021). Molecular Sciences Oxytocin and Bone: Review and Perspectives. https://doi.org/10.3390/ijms22168551

Chen, C., Lanz, R. B., Walkey, C. J., Chang, W., Lu, W., and Johnson, D. L. (2018). Maf1 and Repression of RNA Polymerase III-Mediated Transcription Drive Adipocyte Differentiation. Cell Reports, 24(7), 1852–1864. https://doi.org/10.1016/j.celrep.2018.07.046

Goltzman, D. (2018). Functions of vitamin D in bone. Histochemistry and Cell Biology, 149, 305– 312. https://doi.org/10.1007/s00418-018-1648-y

Goodarzi, H., Nguyen, H. C. B., Zhang, S., Dill, B. D., Molina, H., and Tavazoie, S. F. (2016).

Modulated expression of specific tRNAs drives gene expression and cancer progression.

Cell, 165(6), 1416–1427. https://doi.org/10.1016/j.cell.2016.05.046

Johnson, S. S., Zhang, C., Fromm, J., Willis, I. M., and Johnson, D. L. (2007). Mammalian Maf1 Is a Negative Regulator of Transcription by All Three Nuclear RNA Polymerases. Molecular Cell, 26(3), 367–379. https://doi.org/10.1016/j.molcel.2007.03.021

Liu, P., Ping, Y., Ma, M., Zhang, D., Liu, C., Zaidi, S., Gao, S., Ji, Y., Lou, F., Yu, F., Lu, P., Stachnik, A., Bai, M., Wei, C., Zhang, L., Wang, K., Chen, R., New, M. I., Rowe, D. W., … Zaidi, M. (2016). Anabolic actions of Notch on mature bone. Proceedings of the National Academy of Sciences, 113(15), E2152–E2161. https://doi.org/10.1073/pnas.1603399113

Naot, D., Musson, D. S., and Cornish, J. (2016). The Activity of Adiponectin in Bone. Calcified Tissue International 2016 100:5, 100(5), 486–499. https://doi.org/10.1007/S00223-0160216-5

Orioli, A., Praz, V., Lhôte, P., and Hernandez, N. (2016). Human MAF1 targets and represses active RNA polymerase III genes by preventing recruitment rather than inducing long-term transcriptional arrest. Genome Research, 26(5), 624–634. https://doi.org/10.1101/gr.201400.115

Pagano, A., Castelnuovo, M., Tortelli, F., Ferrari, R., Dieci, G., and Cancedda, R. (2007). New Small Nuclear RNA Gene-Like Transcriptional Units as Sources of Regulatory Transcripts. PLoS Genetics, 3(2), e1. https://doi.org/10.1371/journal.pgen.0030001

Reid, I. R., Baldock, P. A., and Cornish, J. (2018). Effects of Leptin on the Skeleton. Endocrine Reviews, 39(6), 938–959. https://doi.org/10.1210/er.2017-00226

Wein, M. N., and Kronenberg, H. M. (2018). Regulation of Bone Remodeling by Parathyroid Hormone. Cold Spring Harbor Perspectives in Medicine, 8(8).

https://doi.org/10.1101/CSHPERSPECT.A031237